# A single-cell atlas of the testicular interstitium defines Leydig progenitor networks sustaining Leydig cell homeostasis across the lifespan

Xiaojia Huang[1,2,3†], Kai Xia[2†], Meiling Yang[3,4†], Mengzhi Hong[5], Meihua Jiang[2], Weiqiang Li[2], Zhenmin Lei[6], Andy Peng Xiang[2]*, Wei Zhao[2]*

[1]Department of Sports Medicine, Sun Yat-Sen Memorial Hospital, Sun Yat-sen University, Guangzhou, China; [2]Center for Stem Cell Biology and Tissue Engineering, Key Laboratory for Stem Cells and Tissue Engineering, Ministry of Education, Sun Yat-sen University, Guangzhou, China; [3]Medical Research Institute, Guangdong Provincial People's Hospital (Guangdong Academy of Medical Sciences), Southern Medical University, Guangzhou, China; [4]Central Laboratory, The Second Affiliated Hospital of Fujian Medical University, Quanzhou, China; [5]Department of Laboratory Medicine, The First Affiliated Hospital of Sun Yat-sen University, Guangzhou, China; [6]Department of OB/GYN and Women's Health, University of Louisville School of Medicine, Louisville, United States

*For correspondence:
xiangp@mail.sysu.edu.cn (APX);
zhaowei23@mail.sysu.edu.cn
(WZ)

†These authors contributed
equally to this work

Competing interest: The authors
declare that no competing
interests exist.

Reviewing Editor: Lynne-Marie
Postovit, Queen's University,
Kingston, Canada

**Abstract** Declining rates of male fertility pose a significant clinical challenge, while the mechanisms underlying testicular interstitial function remain incompletely understood. Here, we conducted a comprehensive analysis of the single-cell transcriptomic landscape of the murine testicular interstitium across the postnatal lifespan. The investigation unveiled a previously unrecognized population of Cd34+/Sox4+ mesenchymal cells nestled within the interstitium, hinting at their potential as Leydig cell (LC) progenitors. With the aging process of Cd34+/Sox4+ mesenchymal cells, we observed a decline in glutathione levels within the testicular interstitium. Remarkably, these Cd34+/Sox4+ mesenchymal cells exhibited clonogenic self-renewal capacity and a robust propensity to differentiate into LCs. Intriguingly, when transplanted into LC-disrupted or failure models, Cd34+/Sox4+ cells efficiently colonized the testicular interstitium, resulting in a notable increase in testosterone production. Exploring the epigenetic landscape, we identified critical transcription factors, most notably Sox4, governing the stem cell fate of Cd34+/Sox4+ mesenchymal cells. Overall, this comprehensive lifespan-resolved single-cell atlas of testicular interstitial cells provides fundamental insights into LC progenitor biology and regenerative capacity during aging.

## Editor's evaluation

This Important study presents the dynamics and diversity of Leydig cells (LC) across the lifespan, in particular the delineation of Cd34-MC1 as a progenitor source for LC maintenance and regeneration. The evidence supporting the claims is solid and this work will be of interest to investigators in the field of male reproduction.

## Introduction

Declining male fertility has become a significant social and medical concern, with male factors contributing to roughly half of infertility cases. A considerable proportion of male infertility is linked to dysfunctions in the testicular interstitium. Among interstitial cells, Leydig cells (LCs) are crucial for male reproductive function, as they produce testosterone essential for male sexual development and fertility (*Ge et al., 2008*; *Haider, 2004*; *Potter and DeFalco, 2017*; *Zirkin and Papadopoulos, 2018*). During fetal development, LCs drive the masculinization of the reproductive tract and external genitalia, and in adulthood, they maintain male secondary sexual characteristics and support spermatogenesis (*Forest, 1983*). Despite their importance, our understanding of LC development, especially the maintenance of the adult LC population during aging, remains incomplete.

It is well established that two waves of LCs arise in the testis: fetal Leydig cells (FLCs) and adult Leydig cells (ALCs) (*DeFalco et al., 2011*). FLCs emerge during fetal testis development and largely regress after birth, with only a small subset persisting into adulthood (*Liu et al., 2016*; *Shima et al., 2015*; *Shima et al., 2018*; *Wen et al., 2016*). ALCs differentiate at puberty and are responsible for sustaining testosterone production throughout adult life. However, ALCs are generally considered terminally differentiated and have a limited capacity for proliferation or regeneration (*Chamindrani Mendis-Handagama and Siril Ariyaratne, 2001*; *Midzak et al., 2009*). This makes the adult LC population susceptible to age-related decline and dysfunction. Elucidating how the LC pool is maintained or replenished under physiological and aging conditions is therefore of great interest for understanding and potentially treating age-related hypogonadism.

Previous studies have suggested the existence of stem or progenitor cells that give rise to new LCs during postnatal life. For instance, Nestin-expressing cells in the peritubular region were identified as putative stem LCs in the adult testis (*Jiang et al., 2014*; *Zang et al., 2017*). Similarly, other markers have been associated with progenitor-like interstitial cells, such as CD51 (integrin α_v) and Tcf21, indicating that a reserve population capable of LC differentiation might reside in the testicular interstitium (*Shen et al., 2021*; *Zang et al., 2017*). Nonetheless, the exact identity and characteristics of an adult LC progenitor population have remained elusive. Recent advances in single-cell RNA sequencing (scRNA-seq) provide an opportunity to resolve cellular heterogeneity and identify rare cell types in the testis, which could shed light on such progenitor cells and their behavior across different ages.

In this study, we performed comprehensive single-cell transcriptomic profiling of the mouse testicular interstitium at key postnatal stages (neonatal, adolescent, adult, middle-aged, and aged) to investigate how Leydig progenitor populations sustain LC homeostasis across the lifespan. Our analysis uncovered a previously unrecognized population of Cd34$^+$/Sox4$^+$ mesenchymal progenitors, which represent a key component of this regenerative network. These Cd34$^+$/Sox4$^+$ mesenchymal cells are enriched in early postnatal testes and decline in abundance with age, suggesting a developmental regulation of this cell population. The observations demonstrate that they possess clonogenic self-renewal capacity and can differentiate into functional LCs in vitro. In vivo, transplantation of Cd34$^+$/Sox4$^+$ cells into LC-depleted adult testes resulted in efficient engraftment and an increase in testosterone-producing LCs, underscoring their regenerative potential. Subsequent epigenomic analyses identify Sox4 as a critical transcription factor maintaining the stem-like state of these cells. Together, these findings define Cd34$^+$/Sox4$^+$ mesenchymal progenitors as an integral component of the broader Leydig progenitor network that collectively maintains LC regeneration and homeostasis throughout life. This insight provides a mechanistic framework for understanding age-related LC decline and lays the groundwork for developing cell-based therapies for testosterone deficiency and male infertility.

## Results

### Single-cell transcriptomic analysis reveals LC heterogeneity in testicular interstitium

To comprehensively characterize interstitial cell populations within mouse testes, we employed flow cytometry to enrich interstitial cell fractions, maximizing their representation (*Figure 1—figure supplement 1A*). Samples were collected at pivotal developmental stages: neonatal (1 week), adolescent (1 month), adult (2 months), middle-aged (8 months), and aged (24 months). scRNA-seq was performed on these samples utilizing the 10x Genomics Chromium platform (*Figure 1A*). Through

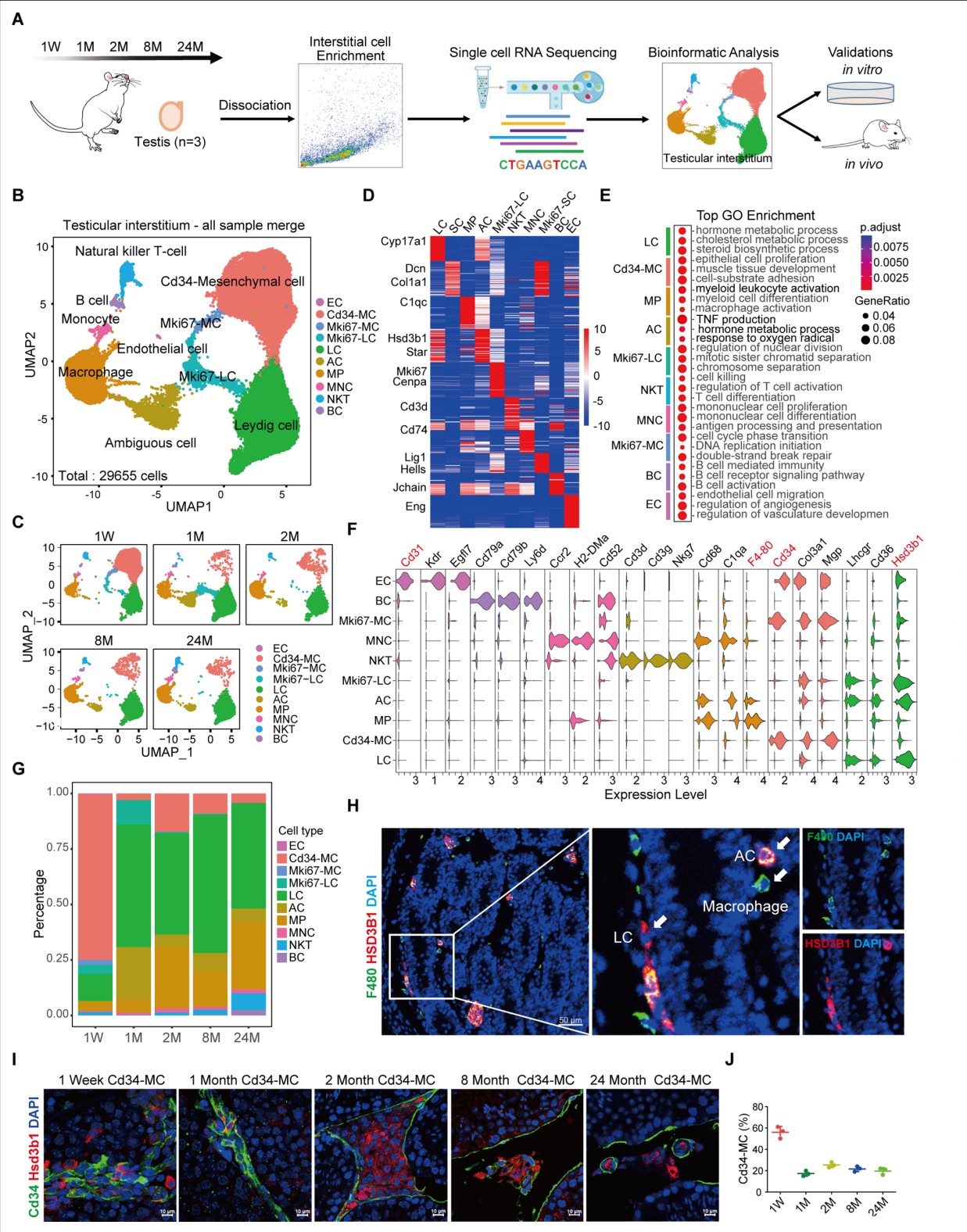

**Figure 1.** Characterization of cell populations in the mouse testicular interstitium. (**A**) Schematic overview of the experimental design for single-cell RNA sequencing conducted on mouse testicular interstitium samples across multiple developmental stages (neonatal, adolescent, adult, middle-aged, and aged), with three replicates at each stage. (**B**) Uniform manifold approximation and projection (UMAP) visualization illustrating the heterogeneity and distribution of distinct cell types within the testicular interstitium across developmental stages. (**C**) UMAP visualization displays various developmental phases: neonatal (1 week), adolescent (1 month), adult (2 months), middle-aged (8 months), and aged (24 months). (**D**) Heatmap displaying the top 10

*Figure 1 continued on next page*

*Figure 1 continued*

differentially expressed genes (DEGs) for each identified cell cluster. (**E**) Dot plot summarizing enriched Gene Ontology (GO) terms corresponding to each cell cluster, highlighting their distinct biological functions. (**F**) Violin plots demonstrating expression patterns of representative marker genes used to identify key interstitial cell populations. (**G**) Quantification of cell type proportions within the testicular interstitium across the different developmental stages. (**H**) Immunofluorescence staining revealing a subpopulation of ambiguous cells (ACs) characterized by co-expression of LC marker, Hsd3b1, and macrophage marker, F4-80, indicated by yellow fluorescence. Scale bar: 50 μm. (**I**) Immunofluorescence labeling for Hsd3b1 and Cd34 to distinguish LCs from MCs within the 2-week-old testicular interstitium. Scale bar: 50 μm. (**J**) Quantitative analysis of Cd34$^+$ MCs within the testicular interstitium across different developmental stages.

The online version of this article includes the following figure supplement(s) for figure 1:

**Figure supplement 1.** Single-cell transcriptome profiling of the mouse testicular interstitium.

**Figure supplement 2.** Dynamic functional changes of testicular macrophages across the postnatal lifespan.

uniform manifold approximation and projection (UMAP) analysis, we identified five major cell types, each defined by hallmark gene expression profiles from existing literature: spermatogenic cells (*Prm1*, *Oaz3*, *Dbil5*), Sertoli cells (*Wt1*, *Sox9*, *Amh*), immune cells (*Cd45*, *Cd68*, *Cd74*), mesenchymal cells (*Dcn*, *Col1a2*, *Col3a1*), and LCs (*Nr5a1*, *Cyp17a1*, *Hsd3b1*) (*Figure 1—figure supplement 1B, C*).

Detailed analysis revealed ten distinct interstitial subpopulations, encompassing well-defined cell types such as LCs, mesenchymal cells (MCs), endothelial cells (ECs), and multiple immune subsets including macrophages (MPs), monocytes (MNCs), natural killer T cells (NKTCs), and B lymphocytes (BCs) (*Figure 1B, C*; *Figure 1—figure supplement 1D, E*). Additionally, unsupervised clustering characterized two proliferative subpopulations: Mki67-LCs, exhibiting high expression of mitotic markers (*Mki67*, *Cenpa*) alongside classical LC markers (*Hsd3b1*, *Star*), and Mki67-MCs, defined by DNA replication-associated genes (*Hells*, *Lig1*) and mesenchymal markers (*Dcn*, *Col1a2*) (*Figure 1B–D*).

Notably, unsupervised clustering also identified an unexpected cell population exhibiting a mixed gene expression signature of both macrophages and LCs, designated as ambiguous cells (ACs) (*Figure 1D, F*). Gene Ontology (GO) analysis of AC-specific differentially expressed genes (DEGs) highlighted involvement in tumor necrosis factor production, hormone metabolism, and response to reactive oxygen species (*Figure 1E*). The AC population showed rapid expansion during adolescence, followed by a significant decline and stabilization at lower proportions in adulthood and aging (*Figure 1G*). Comparative transcriptomic analysis revealed AC-specific upregulation of DEGs associated with responses to external stimuli relative to LCs (*Figure 1—figure supplement 1F, G*), as well as enrichment of hormone metabolism pathways compared to macrophages (*Figure 1—figure supplement 1H, I*).

The single-cell profiling observed significant age-related increases in immune cell populations, notably macrophages, B lymphocytes, and T lymphocytes (*Figure 1G*). Additionally, the analysis uncovered a previously undescribed mesenchymal subpopulation, Cd34-MCs, prominently present during neonatal stages and progressively declining with maturation (*Figure 1G*). Further immunofluorescence staining validated the presence and localization of ACs within the testicular interstitium (*Figure 1H*) and confirmed the sharp reduction of Cd34-MCs post-puberty (*Figure 1I, J*).

To further elucidate immune cell dynamics, we investigated temporal transcriptional changes across immune populations from neonatal to aged stages (*Figure 1—figure supplement 2A*). Macrophages, notably abundant within the adult interstitium, exhibited significant transcriptional shifts throughout development (*Figure 1—figure supplement 2B–D*). CellChat analyses identified consistent macrophage interactions with LCs and Cd34-MCs via the Vcam signaling pathway at all examined stages, suggesting Vcam's role in facilitating macrophage adhesion or migration into testicular tissue. Notably, Vcam pathway signaling intensity markedly decreased by 24 months. Additionally, macrophages were found to secrete factors including Igf1 and Visfatin (*Figure 1—figure supplement 2E*), potentially enhancing steroidogenic activity within LCs (*Hameed et al., 2012*; *Jeremy et al., 2017*; *Lin et al., 1986*; *Ocón-Grove et al., 2010*).

## Dynamic gene expression patterns in interstitial cells from infancy to adulthood

Clustering analysis revealed continuous groupings among LCs, ACs, and MCs, prompting a focused re-clustering of LCs and MCs across developmental stages. This refined clustering yielded seven distinct subpopulations, consistent with previously established research and marker genes:

Cd34-MCs, Mki67-positive MCs (Mki67-MC), progenitor LCs (PLC; *Hsd3b6*, *Cyp51*), immature LCs (ILC; *Cyp17a1*), mature LCs (MLC; *Hsd3b1*), Mki67-positive LCs (Mki67-LC), and AC (*Figure 2A*, *Figure 2—figure supplement 1A*). Additional novel markers specific to each cell type were identified, including *Lcn2* and *Ptgds* for ILCs, *Cyp51* and *Ass1* for PLCs, and *Kcnk3* and *Agt* for MLCs (*Figure 2B*). The heatmap illustrates the top 10 DEGs and representative markers for each cell cluster (*Figure 2—figure supplement 1B*). GO analysis of these DEGs provided functional insights, highlighting their involvement in diverse cellular processes (*Figure 2—figure supplement 1C*). Examination of LC and MC differentiation trajectories, informed by testosterone biosynthesis gene expression patterns, supported a progressive differentiation pathway, suggesting that LCs likely originate from MCs yet exhibit distinct functional properties (*Figure 2—figure supplement 1D, E*). Furthermore, we validated key transcriptomic distinctions at the protein level, confirming differential expression patterns among Mki67-LCs, ACs, PLCs, and MLCs (*Figure 2—figure supplement 1F, G*).

To further elucidate the developmental relationships and differentiation trajectories among LC populations, we conducted pseudotime trajectory analysis across different developmental stages. Significant variations in gene expression profiles were observed between early postnatal (1-week-old) and adult stages. The Cd34-MC population diverged into three distinct subtypes (Cd34-MC1, Cd34-MC2, and Cd34-MC3) with increasing age (*Figure 2C, D*; *Figure 2—figure supplement 1H*). Comparative analysis of their transcriptional profiles highlighted enrichment of pathways related to stem cell functions, including male gonadal development, mesenchymal development, and regenerative processes, particularly within the Cd34-MC1 subtype (*Figure 2E*). Notably, specific markers such as *Sox4* exhibited prominent expression in adolescent Cd34-MC1 cells (Cd34$^+$/Sox4$^+$), indicating their enhanced differentiation potential compared to other subtypes (*Figure 2F*). Immunofluorescence staining at 3 weeks of age confirmed the presence of Cd34$^+$/Sox4$^+$ cells (*Figure 2G*), consistent with Cd34-MC1 identity at this peri-pubertal transitional stage.

To clarify whether Cd34$^+$/Sox4$^+$ cells are SLCs, we analyzed the expression patterns of classical SLC markers across different LC subtypes. The results demonstrated that SLC markers including *Arx1*, *Nr2f2*, *Pdgfra*, and *Tcf21* were all highly expressed in Cd34$^+$/Sox4$^+$ cells (*Figure 2—figure supplement 2A*). Notably, these SLC markers were almost exclusively detected in Cd34-MC1 within the testes of 1-week-old juvenile mice (*Figure 2—figure supplement 2B*). Furthermore, based on the classification of the Cd34-MC subpopulation, the newly identified stem/progenitor cell markers such as *Sox4*, *Mest*, *Wnt5a*, *Ptn*, and *Mdk* exhibit greater specificity compared to SLC markers like *Arx1*, *Nr2f2*, *Pdgfra*, and *Tcf21* (*Figure 2—figure supplement 2C, D*).

Additionally, analysis of publicly available scRNA-seq data from human testes (GSE124263) identified a corresponding CD34-MC subtype in both infant and adult human testicular tissues (*Figure 2H*; *Figure 2—figure supplement 2E, F*). Pseudotime trajectory analysis suggested a conserved progenitor role for human CD34-MCs in LC lineage differentiation (*Figure 2I*; *Figure 2—figure supplement 2G, H*). Remarkably, these CD34-MC subpopulations persisted within the human testicular interstitium from infancy into adulthood, maintaining distinct transcriptional identities throughout development (*Figure 2J, K*).

## Significant reduction in glutathione metabolism within interstitial cells during aging

To characterize age-dependent transcriptional alterations in testicular interstitial cells, we conducted hierarchical clustering of cells (rows) based on the top 2000 most variably expressed genes (columns) across various age groups. This analysis delineated two primary clusters of genes with distinct age-dependent expression profiles. Genes exhibiting elevated expression in older samples included established aging markers such as *Spp1* and *Cxcl10* (*Figure 3A*). Conversely, genes showing decreased expression with aging, such as *Wnt5a* and *Kitl*, formed a distinct cluster potentially linked to protective roles against aging processes (*Figure 3A*). Gene set enrichment analysis (GSEA) demonstrated that genes with reduced expression in older samples were significantly associated with metabolic pathways, particularly cholesterol biosynthesis and steroid hormone metabolism, indicating a potential decline in steroidogenesis during aging (*Figure 3B*). In contrast, genes upregulated with age were prominently involved in immune system pathways, highlighting enhanced activation of both innate and adaptive immune responses within the aging testicular interstitium (*Figure 3C*).

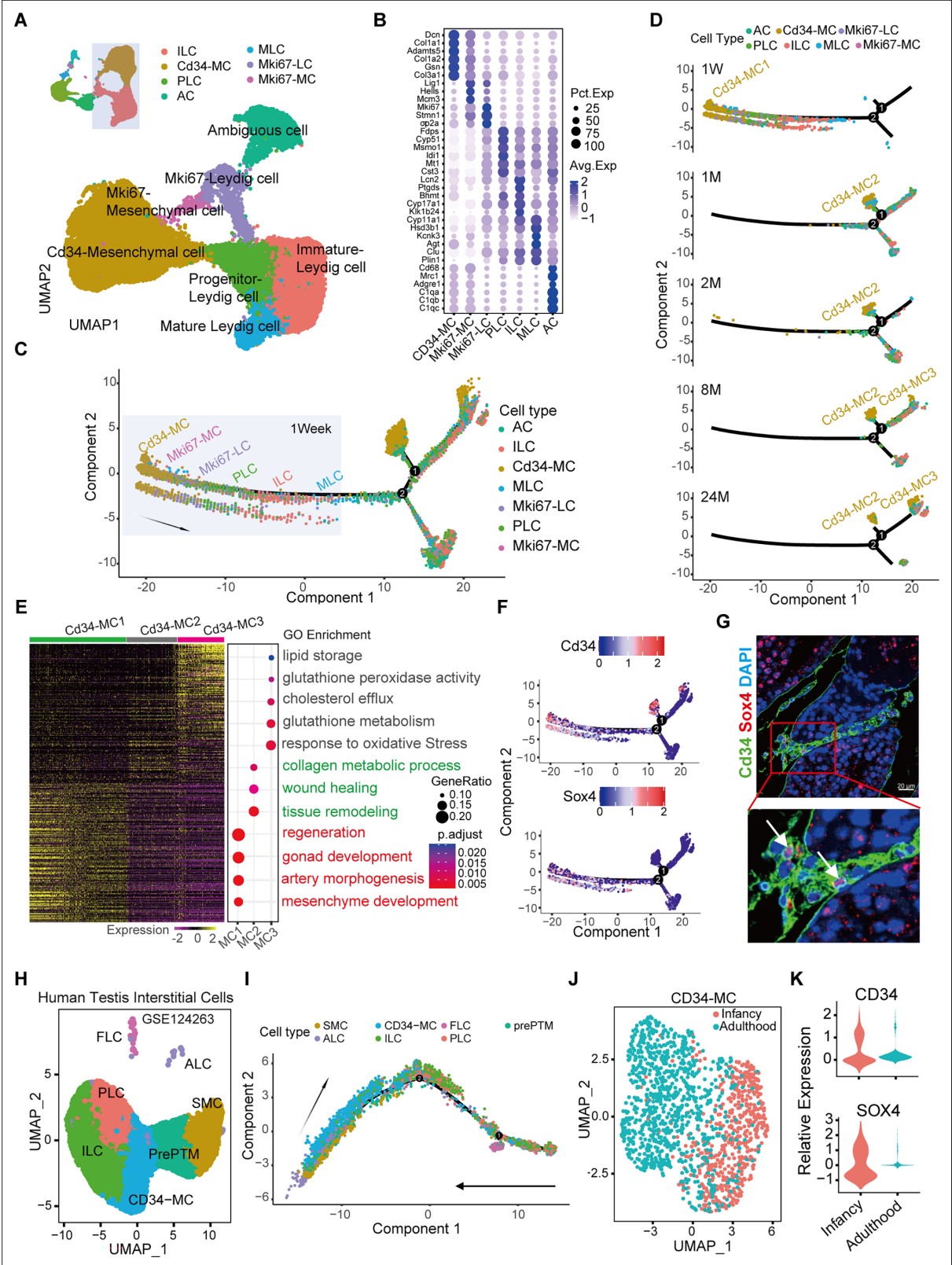

**Figure 2.** Identification and characterization of dynamic Cd34-MC and LC subpopulations across developmental stages. (**A**) Uniform manifold approximation and projection (UMAP) visualization illustrating distinct cell populations within mouse testicular interstitium at various developmental stages (neonatal, adolescent, adult, middle-aged, and aged). (**B**) Dot plot depicting key differentially expressed genes (DEGs) characterizing each Leydig cell subtype. (**C**) Monocle-generated pseudotime trajectory tracing developmental transitions and differentiation paths among Leydig

*Figure 2 continued on next page*

*Figure 2 continued*

cell subtypes. (**D**) Pseudotime analysis revealing dynamic developmental transitions specifically enriched in progenitor Cd34-MC1 cells toward mature Leydig cell lineages. (**E**) Heatmap of top DEGs and their corresponding Gene Ontology (GO) terms enriched within the Cd34-MC subsets. (**F**) Pseudotime plot highlighting gene expression patterns unique to Cd34-MC1 cells. (**G**) Representative immunofluorescence images demonstrating co-localization of Cd34 and Sox4 in interstitial cells at 3 weeks of age. Scale bar: 10 µm. (**H**) UMAP visualization showing the distribution of human interstitial cell populations across infant and adult testicular samples (data sourced from GSE124263). (**I**) Pseudotime trajectory analysis elucidating developmental trajectories of human testicular interstitial cell populations. (**J**) UMAP plots highlighting the distribution of CD34-MC subpopulations in human testes across developmental stages. (**K**) Violin plots illustrating relative gene expression (*z*-score normalized log-transformed values) of indicated genes in Cd34-MC subpopulations.

The online version of this article includes the following figure supplement(s) for figure 2:

**Figure supplement 1.** Subpopulation classification within Leydig cell (LC) and mesenchymal cell (MC) across developmental stages.

**Figure supplement 2.** Molecular characterization of Cd34⁺ mesenchymal cell (MC) subpopulations.

To further investigate metabolic shifts associated with aging, untargeted metabolomics using liquid chromatography–mass spectrometry was performed on testicular tissues from young (3 months) and aged (24 months) mice. Principal component analysis indicated overall metabolic similarities among samples (*Figure 3—figure supplement 1A*). However, volcano plot analysis identified significant metabolic alterations in aging testes, notably elevated levels of peroxidation and hydroxylation products, such as beta-hydroxymyristic acid, D-alpha-hydroxyglutaric acid, and 6-hydroxycaproic acid (*Figure 3—figure supplement 1B, C*). Steroid metabolome profiling further revealed a pronounced decrease in steroid hormones, particularly testosterone, in aged tissues (*Figure 3D*).

Integrating transcriptomic and metabolomic data via MetaboAnalyst 5.0 joint-pathway analysis pinpointed glutathione metabolism as a significantly affected pathway during aging (*Figure 3E*). Consistent with this finding, an age-dependent increase in the expression of glutathione oxidase-related genes has been observed through pseudotime analysis (*Figure 3F*, *Figure 3—figure supplement 1D*), accompanied by a significant decline in reduced glutathione levels (*Figure 3G*). Additionally, interstitial cells from older testes demonstrated elevated expression of genes listed in the GenAge database, including *Gpx4* and *Gsta4*, indicative of heightened oxidative stress responses (*Figure 3—figure supplement 1D and E*).

To identify key transcription factors controlling genes involved in glutathione metabolism, super-enhancer analysis through H3K27 acetylation (H3K27ac) Cut&Tag sequencing has been conducted in Cd34-MCs (*Figure 3H*, *Figure 3—figure supplement 1F*). Notably, super-enhancers associated with Nrf2 were identified within Cd34-MC1 and Cd-MC2 populations but were absent in Cd34-MC3 (*Figure 3I*). Protein–protein interaction network analysis underscored a potential regulatory role of Nrf2 in orchestrating glutathione metabolism-related gene expression (*Figure 3—figure supplement 1G*). Expression analysis revealed consistently high *Nrf2* levels from postnatal to adult stages, followed by a gradual decline in later life (*Figure 3J*). Furthermore, GSEA suggested that reduced *Nrf2* expression in aged LCs could contribute significantly to increased oxidative stress susceptibility (*Figure 3K*).

## Unveiling Cd34-MC1 cells as LC progenitors through transplantation into *Lhcgr*⁻/⁻ mouse models

We next evaluated the self-renewal capacity and differentiation potential of Cd34-MC cells derived from mice at two distinct developmental stages: young (2 weeks old, Cd34-MC1) and aged (24 months old, Cd34-MC3). Cd34-MC1 cells demonstrated significantly enhanced self-renewal abilities and greater clonogenic potential compared to Cd34-MC3 cells isolated from aged mice (*Figure 4—figure supplement 1A*). Furthermore, Cd34-MC1 cells successfully differentiated into LC lineage when cultured in specialized differentiation media, whereas aged Cd34-MC2 cells exhibited limited proliferation and differentiation capacities (*Figure 4—figure supplement 1B*).

To evaluate the in vivo regenerative potential of Cd34-MC1 cells, we transplanted these cells into mice subjected to LC disruption using ethylene dimethanesulfonate (EDS, 200 mg/kg). Cd34-MC1 cells isolated via fluorescence-activated cell sorting (FACS) from 2-week-old mice were transplanted directly into the testicular parenchyma 4 days after EDS-induced LC disruption (*Figure 4—figure supplement 1C*). Serum and testicular tissues were collected at defined intervals (0, 4, 8, 12, 16, and 20 days post-EDS treatment) for subsequent analyses (*Figure 4A*). Remarkably, transplanted Cd34-MC1 cells efficiently localized within the testicular interstitium and contributed substantially to the

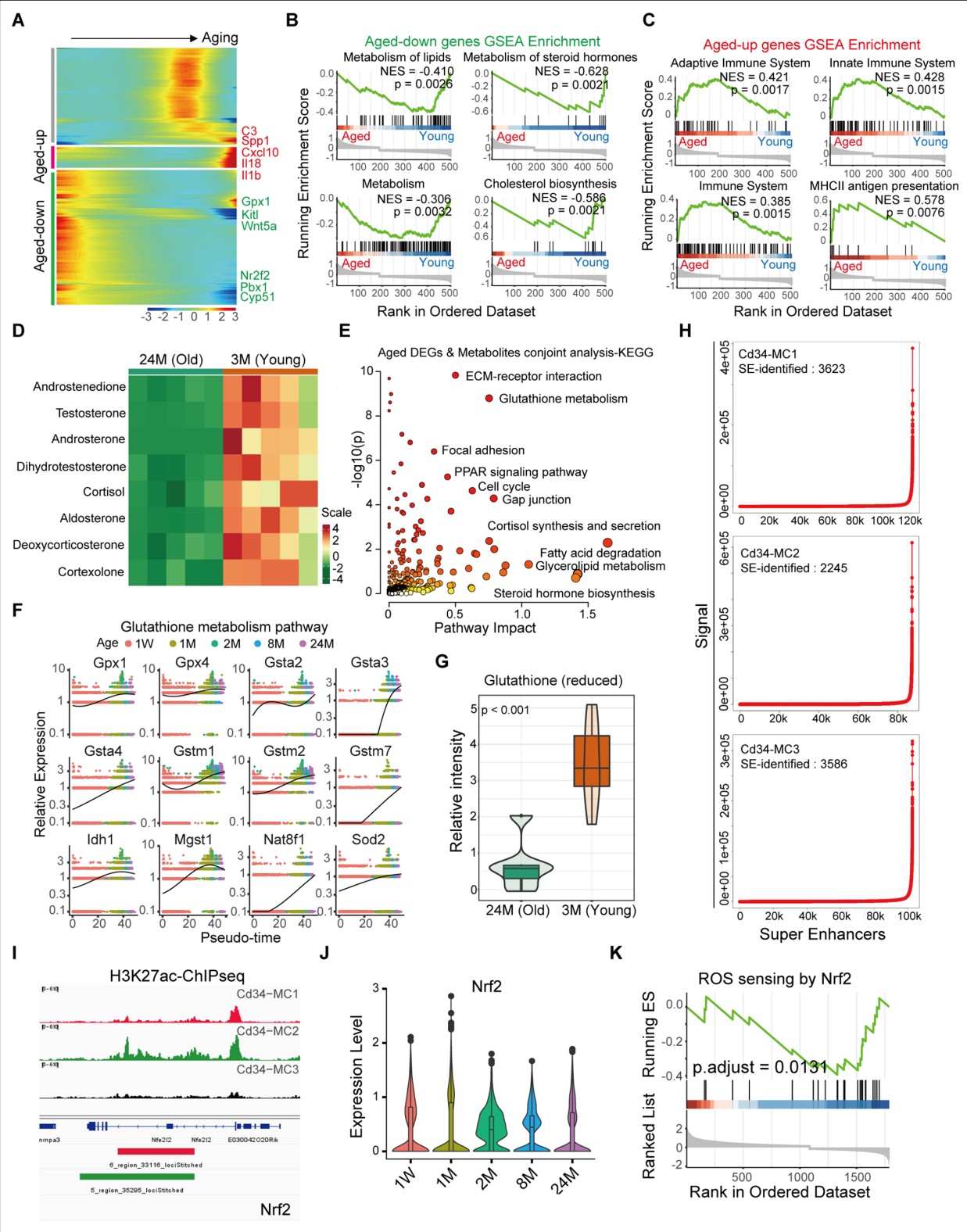

**Figure 3.** Age-associated alterations in glutathione metabolism within testicular interstitial cells. (**A**) *K*-means clustering analysis of genes exhibiting differential expression patterns in testicular interstitial cell populations during aging. (**B**) Gene set enrichment analysis (GSEA) illustrating significantly downregulated metabolic pathways in aged interstitial cells (p < 0.05). (**C**) GSEA showing enrichment of immune-related pathways among genes upregulated in older interstitial cells. (**D**) Targeted steroid hormone metabolomics analysis comparing the testicular lipid profile between young (3-month-old) and aged (24-month-old) mice (*n* = 5). (**E**) Dot plot illustrating correlation between upregulated metabolites and differentially expressed

*Figure 3 continued on next page*

*Figure 3 continued*

genes (DEGs) in aged interstitial cells. (**F**) Expression trends of essential glutathione metabolism pathway genes across developmental stages, highlighting decreased activity in older interstitial cells. (**G**) Relative abundance of reduced glutathione (GSH) quantified in testes from young (3-month-old) and aged (24-month-old) mice ($n = 5$, $p < 0.01$), presented as mean ± SEM. (**H**) A ranked plot of super-enhancers (SEs) showcases those with the highest median H3K27ac scores across Cd34-MC1, Cd34-MC2, and Cd34-MC3 cells. (**I**) Representative ChIP-seq tracks displaying H3K27ac enrichment at the Nrf2 genomic locus in interstitial cells at various ages. (**J**) Violin plots illustrating age-associated changes in Nrf2 gene expression within the Cd34-MC1 cell population. (**K**) GSEA showing enrichment of reactive oxygen species (ROS)-related signaling pathways influenced by Nrf2 activity specifically in aging Cd34-MC1 cells.

The online version of this article includes the following figure supplement(s) for figure 3:

**Figure supplement 1.** Transcriptional changes and metabolic alterations in the testicular interstitium during aging.

---

structural regeneration of LC-depleted testes (*Figure 4B*, *Figure 4—figure supplement 1D*). Notably, transplantation led to a significant increase in Hsd3b1-positive LCs within recipient testes compared to controls (*Figure 4C*, *Figure 4—figure supplement 1E*).

Importantly, Cd34-MC1 transplantation significantly restored serum testosterone levels between days 8 and 16 following EDS treatment (*Figure 4D*). Given testosterone's crucial role in meiosis and spermatogenesis, we analyzed meiotic progression by immunostaining for the meiotic factor Sycp3. A pronounced increase in Sycp3-positive germ cells was observed in testes receiving Cd34-MC1 transplantation compared to the EDS-only controls (*Figure 4—figure supplement 1F, G*). Additionally, progressive sperm motility significantly improved in the transplantation group assessed on day 20 (*Figure 4E*, *Figure 4—figure supplement 1H*).

To further confirm the progenitor role of Cd34-MC1 cells in LC regeneration, we employed lineage-tracing experiments utilizing inducible tdTomato-expressing mice crossed with *Lhcgr* knockout (*Lhcgr*$^{-/-}$) mice, which exhibit impaired LC development. Due to occasional leaky tdTomato expression, Tomato$^{high}$ Cd34-MC cells from testes of both young (MC1) and aged (MC3) tdTomato reporter mice were collected through FACS-based isolation (*Figure 4—figure supplement 1I, J*). These sorted cells were subsequently transplanted into the testes of 1-month-old *Lhcgr*$^{-/-}$ recipient mice (*Figure 4F*). After 3 weeks, a substantial number of tdTomato-positive LCs expressing both Hsd3b1 and Lhcgr were detected in the testes of mice transplanted with Cd34-MC1 cells (*Figure 4G*). In contrast, aged Cd34-MC3 cells failed to significantly contribute to LC regeneration, as evidenced by the absence of tdTomato-labeled *Hsd3b1*-expressing cells in recipient testes (*Figure 4H, I*; *Figure 4—figure supplement 1K, L*). Furthermore, serum testosterone levels significantly increased only in mice receiving Cd34-MC1 cells, underscoring their distinct regenerative and functional superiority compared to aged Cd34-MC3 progenitors (*Figure 4J*). These findings clearly establish Cd34-MC1 cells as potent LC progenitors capable of effective testicular regeneration and functional recovery.

## Identifying Sox4 as a key transcription factor maintaining stemness in Cd34-MC1 cells

H3K27ac ChIP-seq analysis further identified specific super-enhancers associated with critical transcription factors, notably Sox4, within Cd34-MC1 cells (*Figure 5A*). GO analysis revealed that super-enhancers in Cd34-MC1 cells were significantly enriched for terms related to stem cell development (*Figure 5B*). GSEA further underscored Sox4's pivotal role in maintaining the stemness characteristics of Cd34-MC1 cells (*Figure 5C*). Importantly, *Sox4* exhibited elevated expression during the early postnatal stage, which markedly declined throughout development (*Figure 5D*). Consistently, H3K27ac levels at the *Sox4* locus progressively decreased from Cd34-MC1 to Cd34-MC2/3 subpopulations (*Figure 5E*).

To investigate the functional significance of Sox4, we utilized shRNA-mediated knockdown to suppress *Sox4* expression in primary Cd34-MC1 cells (*Figure 5—figure supplement 1A, B*). *Sox4* knockdown significantly increased the proportion of Hsd3b1-positive cells after 48 hr in culture (*Figure 5F*). Quantitative PCR (qPCR) analysis of key steroidogenic pathway genes demonstrated that *Sox4* knockdown notably upregulated expression of *Lhcgr*, *Hsd3b1*, *Star*, *Cyp17a1*, and *Cyp11a1* (*Figure 5G*). Additionally, targeting *Cd34* with shRNA lentivirus resulted in increased expression of *Lhcgr* and *Cyp17a1* but decreased expression of *Hsd3b1*, *Star*, and *Cyp11a1* in Cd34-MC1 cells (*Figure 5—figure supplement 1C–E*). These results suggest that Cd34 is dispensable for maintaining stemness and differentiation potential in these cells.

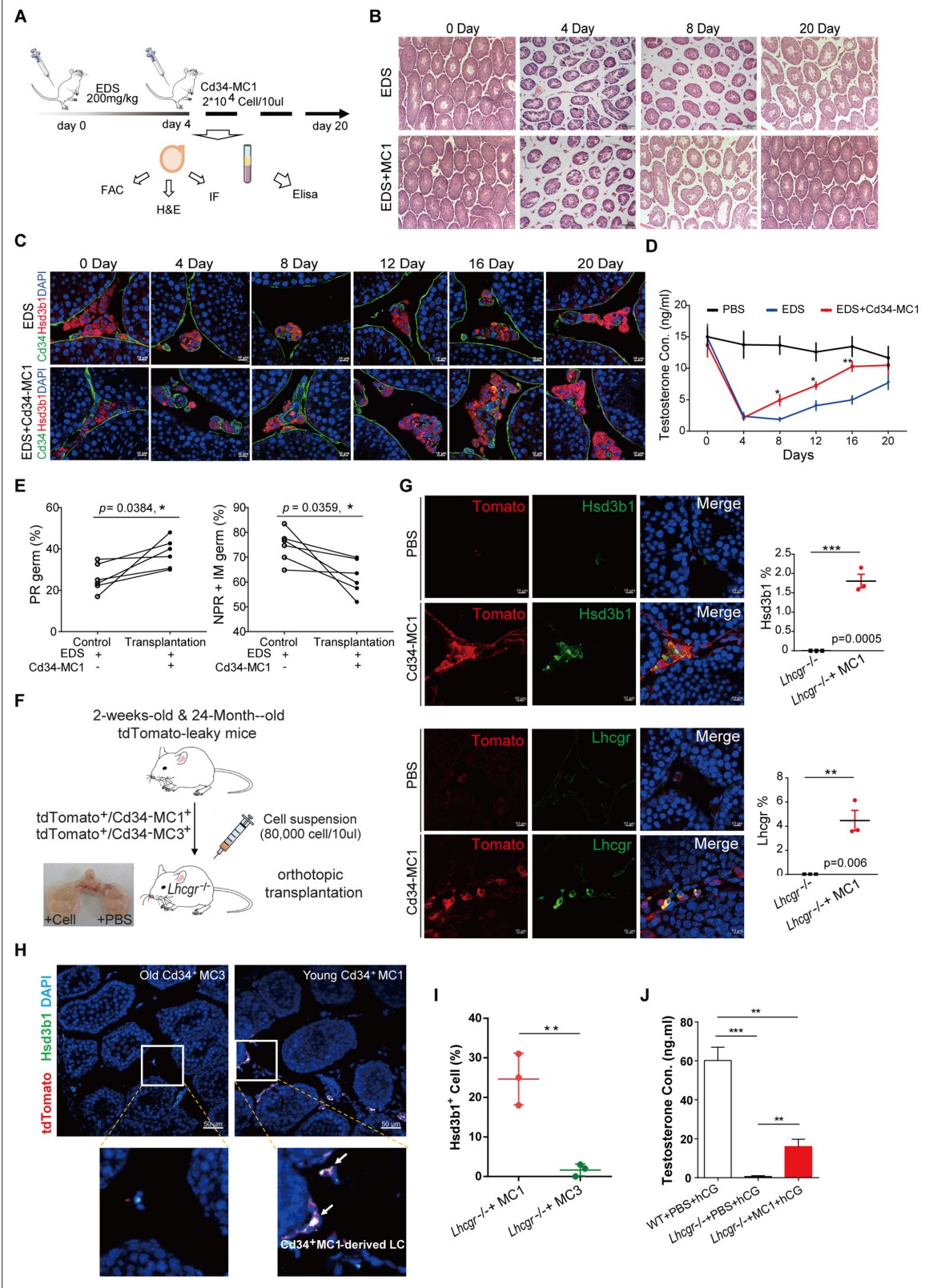

**Figure 4.** Cd34-MC1 cell transplantation enhances testosterone production and Leydig cell regeneration in ethylene dimethanesulfonate (EDS)-treated and aged mice. (**A**) Schematic illustrating the experimental design for Cd34-MC1 cell transplantation in EDS-treated mice. (**B**) Representative histological analysis by hematoxylin and eosin staining of testis sections showing structural restoration post-transplantation of Cd34-MC1 cells. Scale bar: 100 μm. (**C**) Immunofluorescence images confirming an increased presence of Leydig cells marked by Hsd3b1 expression in the testicular

*Figure 4 continued on next page*

*Figure 4 continued*

interstitium following Cd34-MC1 cell transplantation. Scale bar: 50 µm. (**D**) Serum testosterone levels measured at indicated time points post-EDS treatment demonstrating significant restoration following Cd34-MC1 transplantation. Data are presented as mean ± SEM. *p < 0.05 compared to EDS-treated controls, n = 6. (**E**) Improved sperm progressive motility (PR) observed at day 20 post-transplantation compared to controls. Data presented as mean ± SEM. *p < 0.05; **p < 0.01 compared to EDS-treated group, n = 6. (**F**) Experimental schematic depicting Cd34-MC1-tdTomato cell transplantation into testes of *Lhcgr*$^{-/-}$ mice. (**G**) Immunofluorescence images illustrating successful engraftment and differentiation of Cd34-MC1-tdTomato cells into Lhcgr$^+$ and Hsd3b1$^+$ Leydig cells within the interstitium of *Lhcgr*$^{-/-}$ mouse testes. Data are presented as mean ± SEM. *p < 0.05, n = 3, Scale bar: 50 µm. (**H**) Immunofluorescence comparison of Leydig cell regeneration capability between Cd34-MC1 (2-week-old) and Cd34-MC3 (24-month-old) transplanted cells, indicating a superior regenerative potential of younger progenitor cells. Scale bar: 50 µm. (**I**) Quantitative analysis revealing significantly fewer Hsd3b1-positive Leydig cells derived from transplanted Cd34-MC3 compared to Cd34-MC1 cells in *Lhcgr*$^{-/-}$ mouse testes. Data are presented as mean ± SEM. *p < 0.05, n = 3. (**J**) Statistical analysis comparing serum testosterone levels in Cd34-MC1-transplanted *Lhcgr*$^{-/-}$ mice with or without human chorionic gonadotropin (hCG) treatment. Data are presented as mean ± SEM. *p < 0.05, n = 3.

The online version of this article includes the following figure supplement(s) for figure 4:

**Figure supplement 1.** Restoration of testicular function by transplantation of Cd34-MC1 in Leydig cell (LC)-disrupted mouse models.

To further validate Sox4's role in vivo, we injected *Sox4* knockdown or control Cd34-MC1 cells into 1-month-old *Lhcgr*$^{-/-}$ mice (**Figure 5H**). Three weeks post-injection, a substantial increase in LCs has been observed, identified by Hsd3b1 expression, within the testicular interstitium in the *Sox4* knockdown group (**Figure 5I**). Collectively, these findings highlight Sox4 as an essential transcription factor for maintaining stem cell identity in Cd34$^+$/Sox4$^+$ MC1 cells.

## Discussion

The primary goal of this study was to elucidate the cellular dynamics of the testicular interstitium across the male lifespan, with a particular focus on identifying progenitor cells that maintain and regenerate the LC population. Single-cell transcriptomic analysis and complementary functional experiments demonstrated that Cd34$^+$/Sox4$^+$ mesenchymal cells constitute a pivotal progenitor source for LC maintenance and regeneration (**Figure 5J**). Our data show that these cells give rise to adult LCs, and they exhibit remarkable regenerative properties: they form stem cell spheres in culture and readily differentiate into testosterone-producing LCs. When transplanted into models of LC loss or dysfunction, Cd34$^+$/Sox4$^+$ cells efficiently colonized the interstitial niche and restored LC numbers and function. These findings establish Cd34$^+$/Sox4$^+$ mesenchymal cells as a novel stem/progenitor population dedicated to supporting the LC compartment in the testis.

Notably, Cd34$^+$/Sox4$^+$ progenitors predominantly reside in peritubular regions of the testis during the neonatal period, and their frequency diminishes as the organism matures. This observation aligns with the idea that the early postnatal testis harbors a rich pool of stem-like interstitial cells that gradually become depleted or quiescent in adulthood. Previous reports have described CD34-positive interstitial cells in the testis (***Kuroda et al., 2004***), including CD34$^+$PDGFRα$^+$ telocytes that form intricate networks connecting LCs, peritubular myoid cells, and blood vessels (***Abe, 2022***). These telocyte-like cells express adhesion molecules (integrins α4/α9/β1) and VCAM1, suggesting they help create a niche conducive to LC development. However, the precise role of CD34$^+$ cells in adult testis physiology has not been fully clear. Our findings refine this concept by identifying Cd34$^+$/Sox4$^+$ cells as an active subset within this interstitial network that contributes to LC maintenance and regeneration, while exhibiting transcriptional overlap with previously described progenitor populations such as Pdgfra$^+$, Arx$^+$, and Tcf21$^+$ cells. In neonatal testes, they actively contribute to the expanding LC population, and even in adults, a residual subset retains the capacity to regenerate LCs under the right conditions. The involvement of Sox4 as a key transcriptional regulator in these cells is a unique feature of this progenitor population, hinting at specific molecular programs that endow them with stemness and differentiation potential.

The identification of Cd34$^+$/Sox4$^+$ mesenchymal cells thus complements and extends current models of Leydig progenitor biology, integrating them into a broader framework of interstitial progenitors that collectively sustain LC renewal. For example, Nestin-positive interstitial cells were earlier proposed as stem LCs in the rodent testis (***Jiang et al., 2014***), and CD51 (integrin α_v) has been used as a surface marker to isolate stem LCs that can self-renew and give rise to LCs (***Zang et al., 2017***). Transplantation of CD51$^+$ cells in testosterone-deficient animal models resulted in the

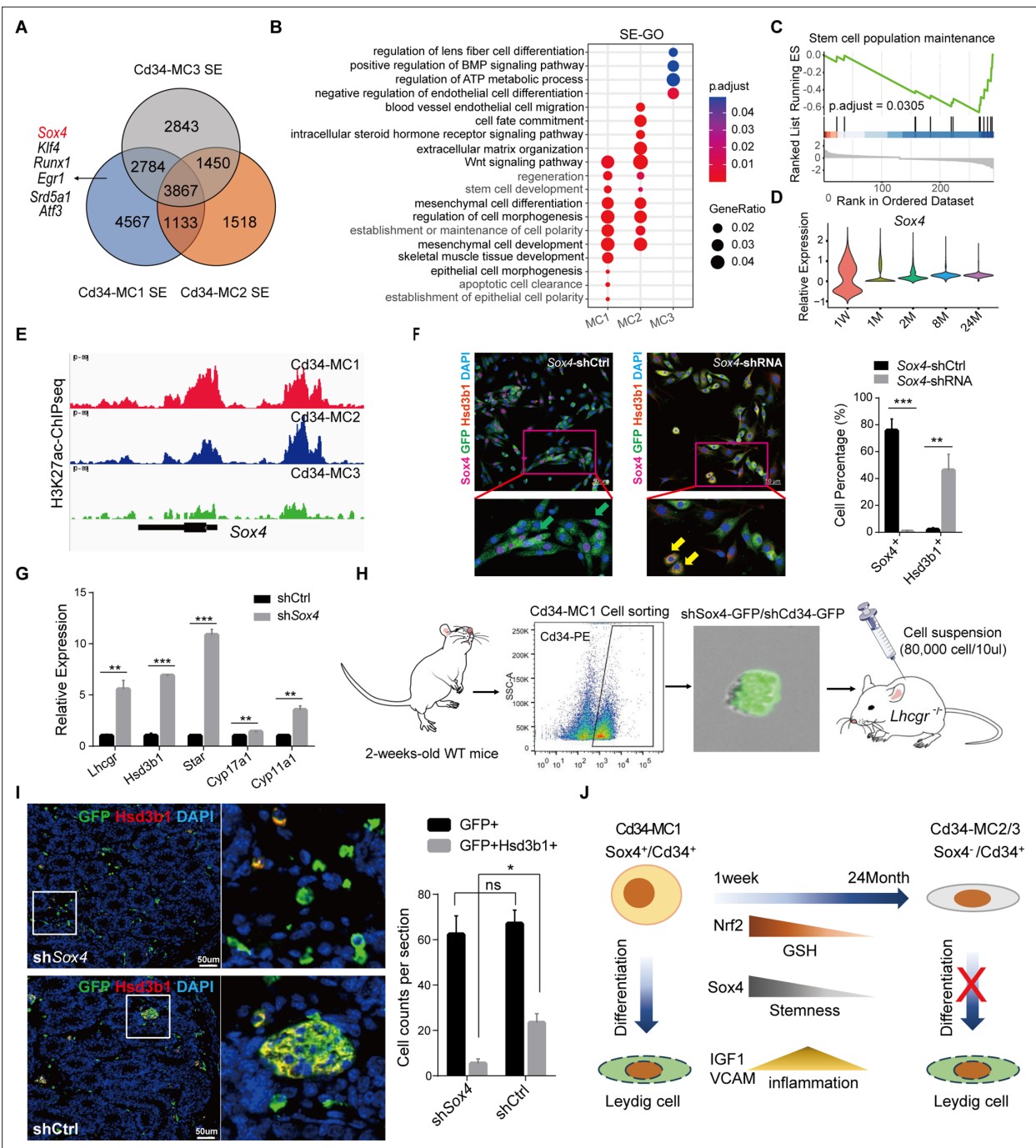

**Figure 5.** SE landscape reveals Sox4 as a crucial transcription factor in maintaining stemness in Cd34-MC1. (**A**) Venn diagram illustrating Cd34-MC1-specific SEs compared with other Cd34-MC subtypes. (**B**) Gene Ontology (GO) enrichment analysis of SE-associated genes in three distinct Cd34-MC subpopulations. (**C**) Gene set enrichment analysis (GSEA) demonstrating significant enrichment of Sox4-regulated genes in pathways essential for maintaining stem cell populations. (**D**) Violin plots showing *Sox4* expression levels across distinct Cd34-MC subsets. (**E**) ChIP-seq profiles highlight H3K27ac mark enrichment at the *Sox4* locus within Cd34-MC1, Cd34-MC2, and Cd34-MC3. (**F**) Immunofluorescence images confirming successful *Sox4* knockdown (green, *Sox4* shRNA) in Cd34-MC1 cells, co-labeled with *Sox4* (purple) and Leydig cell marker Hsd3b1 (red), after 48 hr of shRNA-mediated *Sox4* suppression. Scale bar: 50 µm. Statistical significance indicated as *p < 0.05; **p < 0.01; ***p < 0.001, n = 3. (**G**) qPCR analysis evaluating LC gene expression changes following 48 hr of *Sox4* knockdown in Cd34-MC1 cells. *p < 0.05; **p < 0.01; ***p < 0.001, n = 3. (**H**) Schematic depicting transplantation of Cd34-MC1 cells with or without *Sox4* knockdown into *Lhcgr*−/− mice. (**I**) Immunofluorescence staining demonstrating reduced numbers of Hsd3b1-positive Leydig cells derived from *Sox4*-knockdown *Cd34*-MC1 cells compared to controls in *Lhcgr*−/− mouse testes. Scale bar: 50 µm, n = 3. (**J**) Summary diagram illustrating the key findings of this research.

*Figure 5 continued on next page*

*Figure 5 continued*

The online version of this article includes the following figure supplement(s) for figure 5:

**Figure supplement 1.** Construction of Sox4 and Cd34 knockdown Cd34-MC1.

differentiation of these cells into mature LCs and partial recovery of testicular function (*Zang et al., 2017*). More recently, lineage-tracing studies of *Tcf21*-expressing interstitial cells provided evidence for a bipotential somatic progenitor that can generate both LCs and peritubular myoid cells in adult mice (*Shen et al., 2021*). Within this context, Cd34$^+$/Sox4$^+$ progenitors represent one subset of this coordinated network, defined by a distinctive transcriptional profile (including Sox4 expression) identified through unbiased single-cell profiling. They emerge prominently during early postnatal development and exhibit robust regenerative capacity when transplanted into LC-depleted testes. Their identification and characterization, therefore, enhance rather than replace existing progenitor models and underscore the cooperative nature of multiple progenitor populations in maintaining LC homeostasis. Although FACS-based solely on CD34 enrichment yields a mixed population (~50% purity of Cd34-MC1), our marker validation (Sox4 immunostaining) and consistent regenerative results support Cd34-MC1 as the primary driver of the observed functional effects.

In addition to defining this progenitor population, single-cell atlas captured the broader heterogeneity of interstitial cells and revealed other noteworthy cell types in the testis. For instance, unsupervised clustering identified an AC population co-expressing markers of LCs (e.g., Hsd3b1) and macrophages (F4/80). This hybrid expression profile hints that ACs may represent an intermediate or transitional state with immunological functions in the testis. Interestingly, similar hybrid or fusion-like macrophage-derived cells have been documented in other tissues. In tumors and regenerating liver, macrophages can fuse with epithelial cells, forming hybrids with dual identities. In fibrotic organs, macrophages can transdifferentiate into collagen-producing myofibroblasts (MMT), contributing to tissue remodeling. Notably, testicular macrophages themselves have been shown to express steroidogenic enzymes and participate in androgen biosynthesis. These precedents support the plausibility that macrophages can adopt tissue-specific programs. Thus, the AC population observed here may reflect a functionally specialized, macrophage-derived subset involved in testicular immune–endocrine crosstalk.

We explicitly compared our findings with recent scRNA-seq analyses of testicular aging, such as the study by *Zhang et al., 2023*. Consistent with their observations, we identified aging-associated inflammation, oxidative stress, and reduced steroidogenesis in LCs. Our study significantly expands these findings by analyzing multiple developmental stages across the lifespan and by functionally validating the Cd34$^+$/Sox4$^+$ mesenchymal progenitors, directly linking their decline to impaired LC regeneration during aging.

Finally, integrative analysis shed light on how age-related changes in the testicular microenvironment might influence LC function and the activity of progenitor cells. Single-cell analysis observed signs of increasing oxidative stress in the aging testis, such as reduced antioxidant gene expression in older LCs and lower glutathione levels, consistent with reports that aged LCs produce more reactive oxygen species and have diminished antioxidant defenses. Notably, the antioxidant regulator Nrf2 was found to be under super-enhancer control in Cd34$^+$ progenitor cells, yet its expression declined in mature LCs with age. This suggests that young Cd34$^+$/Sox4$^+$ mesenchymal cells may help maintain a redox-balanced niche which is lost as the cells (and Nrf2 activity) wane over time, potentially contributing to the age-associated drop in testosterone production. In parallel, we found that testicular macrophages likely play a supportive role in LC development early in life but may become detrimental with aging. Prepubertal macrophages secrete factors (e.g., Vcam1, Igf1) that support LC differentiation, whereas aging macrophages adopt an immunosuppressive, pro-inflammatory phenotype and can produce excess ROS. Such changes in macrophages could impair Cd34$^+$/Sox4$^+$ progenitor differentiation and hinder LC steroidogenesis in older animals. Together, these insights emphasize that a healthy metabolic and immune environment is important for optimal function of LCs and their progenitors. We have de-emphasized these metabolic and macrophage findings in our interpretation to keep the spotlight on the progenitor cells; however, they provide important context for how systemic aging might modulate the effectiveness of the LC regenerative process.

In summary, this study provides a high-resolution atlas of the murine testicular interstitium across postnatal development and aging, uncovering a novel Cd34$^+$/Sox4$^+$ mesenchymal cell population that

serves as LC progenitors. By delineating previously unappreciated cell types and their interactions, we highlight the critical importance of cellular diversity in the maintenance of testicular function. The discovery of a dedicated LC progenitor pool opens new avenues for understanding testis aging and for developing regenerative therapies. Our findings not only advance fundamental knowledge of how the adult LC population is sustained but also pave the way toward cell-based interventions for conditions such as age-related hypogonadism and other forms of testosterone deficiency.

## Materials and methods

### Animals

Wild-type C57BL/6J mice, Lhcgr knock-out mice in C57BL/6J background and Rosa26-CAG-loxp-stop-loxp-tdTomato mice were used in this study. The wild-type C57BL/6J mice were purchased from GemPharmatech, China, the Lhcgr knockout mice were donated by Zhenmin Lei's laboratory, and the Rosa26-CAG-loxp-stop-loxp-tdTomato mice were donated by Zhao Meng's laboratory.

### Isolation and capture of mouse testicular interstitial cells

Testis samples from C57BL/6J mice were dissected aseptically, and the tunica albuginea was carefully removed to enrich interstitial cells, as previously described. The interstitial compartment was then gently dissociated using collagenase type IV (1 mg/ml in DMEM-F12, phenol red-free) at 37°C for 10 min, preserving the seminiferous tubule structure intact. During digestion, tissues were gently pipetted up and down to achieve optimal dissociation while maintaining cell viability. Digestion was halted by adding 2% FBS, and the cell suspension was filtered through a 40-μm nylon mesh to remove tissue debris, followed by gentle red blood cell lysis. Cells were centrifuged (350 rcf, 5 min, room temperature) and resuspended in DPBS or 2% FBS in DPBS.

### Flow cytometry analysis and cell sorting

For single-cell transcriptional profiling, the interstitial cells were suspended in DPBS containing 2% FBS and loaded onto a FACSAria III instrument for FACS. After sorting, the interstitial cells were selected for the construction of a single-cell RNA-sequencing library.

To identify markers and for cell culture experiments, testicular cells obtained after collagenase dissection were suspended in DPBS. The cells were incubated with primary antibodies, specifically Cd34-PE /Cd34-FITC and Cd36-APC, at 4°C for 30 min to facilitate marker identification. To minimize fluorescence quenching, the samples were protected from light during this process. The unbound antibodies were removed by washing with DPBS containing 2% FBS. Subsequently, the cells were centrifuged at 350 rcf for 5 min at room temperature and re-suspended in DPBS containing 2% FBS for flow cytometry analysis and cell sorting.

### Clonal sphere formation assay for Cd34-MC1

Cd34$^+$ cells were diluted to a density of 100 cells/ml and seeded into ultra-low adherent 96-well plates at a volume of 10 μl per well. A total of 150 μl of medium was added to each well. The medium consisted of DMEM/F12 supplemented with 5% chicken embryo extract, 1 nM dexamethasone, 1% ITS, 1% non-essential amino acids, 1% N2 and 2% B27 supplements, 0.1 mM β-mercaptoethanol, and 1 ng/ml LIF, 20 ng/ml bFGF, 10 ng/ml EGF, 20 ng/ml PDGF-BB, and oncostatin-M.

Wells containing only one cell were marked, and the cultures were observed daily. Spheres were identified as free-floating spherical structures with a diameter greater than 50 μm. The cultures were maintained in a 5% CO2 water-jacketed incubator at 37°C, and the medium was changed every 3 days.

### Cell differentiation culture

To initiate cell differentiation, 20,000 Cd34$^+$ cells were suspended in a medium and plated on gelatin-coated glass-bottom cell culture dishes. The medium used was composed of phenol red-free DMEM/F12 supplemented with 2% FBS, 10 ng/ml PDGF-BB, 1 ng/ml LH, 1 nM TH, 70 ng/ml IGF1, and 1% ITS supplement.

The cells were incubated at 37°C in a 5% CO2 environment for 3 days. Proliferation of the cells was assessed using the Click-iT Edu imaging kit, following the manufacturer's instructions. Subsequently, differentiation was confirmed by immunostaining for lineage-specific markers of LCs.

## Cd34-MC1 cell transplantation

Ninety male C57BL/6J mice aged 3 months and ten male C57BL/6J mice aged 24 months were included in this study. The 3-month-old mice were randomly divided into three groups: Control group, EDS injury group, and Cell transplantation group (*n* = 5 for each group at each time point). The control group received an intraperitoneal injection of DPBS, while the other two groups received a single dose of 200 mg/kg body weight of EDS, which led to the depletion of LCs in the adult testis within 4–6 days.

To assess whether the injection of Cd34+ cells could facilitate the recovery of LC dysfunction, the Cd34+ cells were washed with DPBS, approximately 20,000 Cd34+ cells in 10 µl of PBS were injected into the parenchyma of the recipient testes 4 days after the 3-month-old mice had received EDS treatment. Testes and serum samples from all animals were collected at 0, 4, 8, 12, 16, and 20 days after EDS treatment for subsequent flow cytometry analysis, histological analysis, and measurement of testosterone concentration.

## Cd34-MC-tdTomato cell transplantation

Cd34-MC-tdTomato cells were isolated from the testes of about 2 weeks (Cd34-MC1) and 24 months (Cd34-MC3) male tdTomato-leaky Rosa26-CAG-loxp-stop-loxp-tdTomato mice using flow cytometry. These isolated cells were then resuspended in DPBS. Subsequently, we performed orthotopic transplantations into the C57BL/6J *Lhcgr*−/− mice. The right testis tissue of each C57BL/6J *Lhcgr*−/− mouse received an injection of 80,000 Cd34-MC-tdTomato cells in a volume of 10 µl per tissue, while the left testis tissue received a control injection of 10 µl of DPBS. After a 3-week period, we collected testes from six of the mice that had undergone cell transplantation for immunofluorescence staining.

## Testosterone concentration assay

At each specified time point of EDS model, mouse sera were collected for quantitative measurement of testosterone levels. To evaluate hypothalamic–pituitary–gonadal axis regulation, mice received an intraperitoneal injection of human chorionic gonadotropin (10 IU) prior to serum collection to stimulate testosterone production. Testosterone concentrations were subsequently measured using a commercially available ELISA kit according to the manufacturer's instructions.

## *Sox4*-shRNA infection and assessment

We constructed shRNA sequences targeting *Sox4* or *Cd34* and integrated them into the pTSBX-U6-shRNA-EF1-copGFP-2A-PURO lentiviral vector. Flow cytometry was utilized to sort Cd34-MC1 cells, which were then infected with lentiviruses carrying specific shRNA for 12 hr. After the infection period, cells were cultured in fresh medium for up to 48 hr. The efficiency of gene silencing was assessed using quantitative PCR (qPCR). The shRNAs and primers used in this paper are provided in *Supplementary file 1*.

## Immunofluorescence of the mouse testis

Immunofluorescence staining was performed on 5 µm formalin-fixed, paraffin-embedded sections from testis samples. Sections were rehydrated and subjected to heat-mediated antigen retrieval in 10 mM sodium citrate buffer (pH 6.0). After blocking with goat serum working solution for 30 min, sections were incubated overnight at 4°C with a mix of diluted primary antibodies at their recommended dilutions. Antigen detection was conducted using Alexa Fluor 488 (Thermo Fisher Scientific, A11034, A11029) and Alexa Fluor 594 (Thermo Fisher Scientific, A11032, A11037) secondary antibodies (1:500) for 2 hr at room temperature in the dark. All primary and secondary antibodies were diluted in 0.5% BSA. After three washes in PBS, sections were counterstained with Hoechst 33342 (1:2000 in PBS) to visualize nuclei. Images were captured using a Zeiss LSM780 Confocal Laser Scanning Microscope with 20x or 63x oil immersion objectives and analyzed using ZEN software. The primary and secondary antibodies used in this study are listed in *Supplementary file 2*.

## Histological examination

Testes were dissected from mice, fixed in 4% paraformaldehyde for 12 hr at 4°C, embedded in paraffin, and sectioned into 5 µm slices. Before staining, tissue sections were dewaxed in xylene, rehydrated through a series of decreasing ethanol concentrations, and washed in distilled water. Sections were

stained with hematoxylin and eosin, dehydrated through increasing ethanol concentrations, cleared in xylene, and mounted with neutral balsam. Stained sections were examined using an Olympus BX63 microscope.

## Cut&Tag library construction

CUT&Tag for profiling histone H3K27ac was performed using the Hyperactive Universal CUT&Tag Assay Kit for Illumina (Vazyme, TD903-02), and anti-H3K27ac antibodies were used to enrich target DNA fragments from Cd34-MC1, MC2, and MC3 cells obtained by flow sorting. TruePrep Index Kit for Illumina (v21.2, Vazyme, TD202) was used for DNA library construction.

## Data analysis of Cut&Tag

For H3K27ac-CUT&Tag data, clean reads were aligned to the *Mus musculus* reference genome (mm9) using Bowtie2 software (v2.4.0). Duplicated reads were removed, and only uniquely mapping reads were retained for further analysis. MACS2 (v2.1.1) was used to call peaks using the parameters -p 0.05. For broad peaks, H3K27ac signal intensity from CUT&Tag data was used to define SEs. SEs were identified and annotated using the default parameters -t 2500, following the ROSE algorithm (*Whyte et al., 2013*). BigWig files were visualized using IGV (v2.16.0), and heatmaps and signal profiles were generated using deepTools.

## Quantitative real-time PCR

Total RNA was extracted from the cells using TRIzol reagent, following the manufacturer's instructions. The concentration and purity of RNA were assessed using a NanoDrop spectrophotometer. Complementary DNA (cDNA) was synthesized from 1 μg of total RNA using the High-Capacity cDNA Reverse Transcription Kit. Quantitative real-time PCR was performed using the SYBR Green PCR Master Mix on the QuantStudio 5 Real-Time PCR System. The thermal cycling conditions were 95°C for 10 min, followed by 40 cycles of 95°C for 15 s and 60°C for 1 min. Gene expression levels were normalized to GAPDH and calculated using the $2^{-\Delta\Delta Ct}$ method. Each sample was analyzed in triplicate.

## Processing of scRNA-seq data

scRNA-seq data of testicular samples were aligned and quantified using the Cell Ranger Software Suite (v3.0.2, 10X Genomics). Approximately 7000–120,000 cells per sample were captured. The reads were aligned to the *M. musculus* reference genome (mm10), and after alignment, filtering, barcode counting, and UMI counting using the Cell Ranger pipeline, feature-barcode matrices were generated.

Quality control was performed using the Seurat R package (v4.0) (*Hao et al., 2021*; *Qiu et al., 2017*). Cells expressing fewer than 300 genes or more than 6000 genes as well as more than 40,000 counts were excluded to avoid empty droplets and doublets. Genes expressed in fewer than 20 cells were also removed from analysis. Additionally, cells with >25% mitochondrial gene content were excluded to remove dead or damaged cells.

To further ensure data quality, doublet detection was explicitly performed using the DoubletFinder package and removed according to their barcodes. Only high-quality singlets were retained for subsequent analysis.

To correct for batch effects, we used the FindIntegrationAnchors function in Seurat (v4.0) (*Stuart et al., 2019*) to integrate data across different samples. We also applied another batch correction method, Harmony, to correct for batch effects and obtain consistent clustering results. The FindClusters function with a resolution of 0.25 was used to identify 17 distinct cell clusters, and FindMarkers was used to determine DEGs and identify marker genes for each cell type.

## Cell type identification and definition

Single-cell transcriptomic analysis enabled the identification of distinct interstitial cell populations:

Mesenchymal cells (MCs) were defined as multipotent stromal interstitial cells, following classical descriptions (*Roberts and Skinner, 1990*), further characterized as stromal niche-supporting cells (*Kfoury and Scadden, 2015*).

LCs were identified by their steroidogenic marker expression and classified based on developmental lineage definitions established previously (*Chen et al., 2009*).

Progenitor-like cells (PLCs) were defined as early Leydig progenitors, exhibiting partial steroidogenic gene expression and proliferative capacity, consistent with prior descriptions of Leydig progenitors or stem LCs (*Jiang et al., 2014*; *Shen et al., 2021*; *Shen et al., 2021*; *Chikhovskaya et al., 2014*).

### Gene enrichment analysis

To identify the functions and crucial pathways of DEGs across the different cell types, the DEGs were analyzed by the ClusterProfiler package (v4.0.5) (*Wu et al., 2021*; *Yu et al., 2012*) through the GO, Kyoto Encyclopedia of Genes and Genomes, and GSEA databases. All the significantly enriched terms were determined using the p-value cutoff = 0.05, while the enrichment of GSEA terms also shows the normalized enrichment score, indicating statistical significance.

### Trajectory analysis

To investigate the temporal progression and developmental trajectories of LC and mesenchymal cell subtypes, we performed pseudotime analysis using Monocle (v2.20.0) (*Trapnell et al., 2014*). This analysis was conducted based on the expression patterns of highly variable genes that were identified by Seurat. Pseudotime represents a continuous trajectory that captures the inferred developmental progression of individual cells within a population.

### Definition of functional scores

In order to evaluate the functional characteristics of testicular macrophages at different stages of postnatal development, we employed the 'AddModuleScore' function within Seurat to calculate functional scores for specific gene sets of interest. These gene sets were chosen based on their relevance to macrophage biology and included the following functional categories: macrophage migration (GO:1905517), antigen processing and presentation (GO:0019882), phagocytosis (GO:0006909), and macrophage cytokine production (GO:0010934). Additionally, we incorporated M1/M2 polarization gene sets, which were obtained from a previously published study (*Sun et al., 2021*).

### Cell–cell interaction analysis

To investigate cell–cell interactions within the mouse testis, we utilized the CellChat software (v1.1.2) (*Jin et al., 2021*). Several functions within CellChat were employed to analyze and visualize the signaling pathways involved in cell–cell communication. First, we utilized the 'netVisual_chord_gene' function to visualize the signaling pathways originating from macrophages and targeting LCs or Cd34+ mesenchymal cells. This allowed us to explore the interactions and potential signaling events between these cell types. Next, the 'netVisual_aggregate' function was employed to display selected signaling networks related to the signaling pathways of interest. This function enabled us to focus on specific signaling pathways and investigate the interactions between different cell populations within the testis. Furthermore, the 'netVisual_bubble' function was utilized to present specific ligand–receptor pairs associated with the signaling pathways of interest. Ligand–receptor pairs such as VCAM, IGF, VISFATIN, and GAS signaling pathways were specifically examined, providing insights into the potential molecular interactions and communication between cells involved in these pathways.

### Statistical analysis

Quantitative results are presented as mean ± SEM. ANOVA was used for multiple comparisons; where significance was detected, post hoc testing was then carried out (GraphPad Prism). All other quantitative analyses were analyzed using unpaired Student's *t*-tests. Statistical analyses are indicated in figure legends. Significant outliers were detected using Grubb's test (GraphPad Prism). Significance was determined at p < 0.05.

### Ethical approval statement

All mice used were housed and cared for in accordance with ethical guidelines, and experimental protocols were approved by the Ethics Committee of Sun Yat-sen University (No. 2017-133).

## Acknowledgements

This work was supported by the National Key Research and Development Program of China (2023YFC2506100 and 2017YFA0103800) and National Natural Science Foundation of China (82172698 and 82303201). The High-level Hospital Construction Project (DFJHBF202102).

## Additional information

### Funding

| Funder | Grant reference number | Author |
| --- | --- | --- |
| National Key Research and Development Program of China | 2023YFC2506100 | Wei Zhao |
| National Natural Science Foundation of China | 82303201 | Meiling Yang |
| National Key Research and Development Program of China | 2017YFA0103800 | Wei Zhao |
| High-level Hospital Construction Project of Guangdong Provincial People's Hospital | DFJHBF202102 | Wei Zhao |

The funders had no role in study design, data collection, and interpretation, or the decision to submit the work for publication.

### Author contributions

Xiaojia Huang, Data curation, Formal analysis, Validation, Investigation, Visualization, Methodology, Writing – original draft, Project administration, Writing – review and editing; Kai Xia, Supervision, Validation, Writing – review and editing; Meiling Yang, Data curation, Methodology, Writing – original draft; Mengzhi Hong, Data curation, Software, Formal analysis; Meihua Jiang, Weiqiang Li, Supervision, Funding acquisition; Zhenmin Lei, Supervision, Methodology; Andy Peng Xiang, Supervision, Funding acquisition, Project administration, Writing – review and editing; Wei Zhao, Conceptualization, Supervision, Funding acquisition, Validation, Writing – original draft, Project administration, Writing – review and editing

### Author ORCIDs

Xiaojia Huang ⓘ https://orcid.org/0009-0005-1477-1644
Andy Peng Xiang ⓘ https://orcid.org/0000-0003-3409-5012
Wei Zhao ⓘ https://orcid.org/0000-0002-0774-2571

### Ethics

This study was performed in strict accordance with the recommendations in the Guide for the Care and Use of Laboratory Animals. All of the animals were handled according to 'Regulations on the management of experimental animals' and 'Guidelines on the good treatment of laboratory animals' of the Sun Yat-sen University. The protocol was approved by the Committee on the Ethics of Animal Experiments of the Sun Yat-sen University (Permit Number: 2017-133). All surgery was performed under sodium pentobarbital anesthesia, and every effort was made to minimize suffering.

### Decision letter and Author response

Decision letter https://doi.org/10.7554/eLife.100396.sa1
Author response https://doi.org/10.7554/eLife.100396.sa2

## Additional files

### Supplementary files

MDAR checklist

Supplementary file 1. shRNAs and primers.

Supplementary file 2. Primary and secondary antibodies.

Supplementary file 3. Chemicals, Peptides, and Recombinant Proteins.

Supplementary file 4. Software and Algorithms.

Supplementary file 5. Critical Commercial Assays.

Supplementary file 6. Instrument and Equipments.

## Data availability

Sequencing data have been deposited in GEO under accession codes GSE262415 and GSE262416. All data generated or analyzed during this study are included in the manuscript and supplemental information files. No new code has been generated for this manuscript, software used is cited in *Supplementary file 4*.

The following datasets were generated:

| Author(s) | Year | Dataset title | Dataset URL | Database and Identifier |
|---|---|---|---|---|
| Huang X, Xia K, Yang M, Hong M, Jiang M, Li W, Lei Z, Xiang AP, Zhao W | 2024 | A comprehensive atlas of testicular interstitium reveals Cd34+/Sox4+ mesenchymal cells as potential Leydig cell progenitors | https://www.ncbi.nlm.nih.gov/geo/query/acc.cgi?acc=GSE262415 | NCBI Gene Expression Omnibus, GSE262415 |
| Huang X, Xia K, Yang M, Hong M, Jiang M, Li W, Lei Z, Xiang AP, Zhao W | 2024 | A comprehensive atlas of testicular interstitium reveals Cd34+/Sox4+ mesenchymal cells as potential Leydig cell progenitors | https://www.ncbi.nlm.nih.gov/geo/query/acc.cgi?acc=GSE262416 | NCBI Gene Expression Omnibus, GSE262416 |

The following previously published dataset was used:

| Author(s) | Year | Dataset title | Dataset URL | Database and Identifier |
|---|---|---|---|---|
| Sohni A, Tan K, Song HW, Burow D | 2018 | Neonatal and adult human testis defined at the single-cell level | https://www.ncbi.nlm.nih.gov/geo/query/acc.cgi?acc=GSE124263 | NCBI Gene Expression Omnibus, GSE124263 |

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
