## [Editor Report]

This Important study presents the dynamics and diversity of Leydig cells (LC) across the lifespan, in particular the delineation of Cd34-MC1 as a progenitor source for LC maintenance and regeneration. The evidence supporting the claims is solid and this work will be of interest to investigators in the field of male reproduction.

---

## [Decision Letter]

**Decision letter after peer review:**

Thank you for submitting your article "A comprehensive atlas of testicular interstitium reveals Cd34+/Sox4+ mesenchymal cells as potential Leydig cell progenitors" for consideration by *eLife*. Your article has been reviewed by 2 peer reviewers, and the evaluation has been overseen by a Reviewing Editor and Lynne-Marie Postovit as the Senior Editor.

Essential Revisions:

1) The paper should be edited for clarity, and spelling as described in the reviewers reports. In particular, the authors should remove irrelevant information and should articulate the novelty of the work.

2) In terms of data analysis, the author should eliminate batch effects and provide detailed analysis procedures, which are crucial for the reliability of the defined cell subpopulations and the lineage differentiation of LCs. For example, the authors should take care to eliminate contaminating cells in the scRNAseq experiments. Additionally, the presence of negative values for gene expression in most of the figures (e.g., Fig2K) can be addressed by annotating the z-score in the legend or by presenting the readers with normalized gene expression values, as gene expression inherently does not have negative values. The current annotation method in the text is prone to mislead readers.

3) The authors should refer to the classical evaluation standards of SLC transplantation experiments and use more comprehensive data to support the findings of this paper. Additionally, knockdown experiments should include analysis at the protein level (supplementary Fig6), and further verification is needed to check whether the results from immunostaining are consistent with the statistics (e.g. Fig1H).

4) The CD34+ population should be better described and more specific markers should be added to distinguish PLC,ILC and MLC.

5) All antibodies (for example against HSD3B1) should be validated with positive and negative controls to ensure specificity.

*Reviewer #1 (Recommendations for the authors):*

Huang et al., conducted a comprehensive analysis of the sc-transcriptomic landscape of the murine testicular interstitium across the postnatal lifespan. Moreover, they proposed Cd34-MC1 as a potential Leydig cell progenitor. In detail, these cells exhibited self-renewal capacity and potential to differentiate into LC. Notably, upon being transplanted into EDS or Lhcgr-/- models, Cd34-MC1 could colonize the testicular interstitium, generating an increase in testosterone production. They further identified Sox4 in maintaining the stemness of Cd34-MC1. Besides, a decline in glutathione levels within the testicular interstitium was observed during the aging process. Altogether, this work broadens the understanding of a possible progenitor source for LC, which may provide a new tool for the studies of testosterone deficiency treatment.

However, many conclusions in this study are not well supported by sufficient evidence, and more rigor should be applied to data analysis and experimental design.

1. There are several issues in data analysis that affect cell definition and specificity of developmental stage, leading to less reliable conclusions regarding LC lifespan: 1) the definition of cell identities lacks supporting evidence from literature, for instance, mesenchymal cell and PLC; 2) the authors have not clarified whether they have excluded doublets from the analysis and whether the AC subpopulation still exists after the removal of doublets? The low number of ACs observed in Fig1G raises this concern; 3) regarding the analysis of LC within the lifespan, the pseudo-time trajectory presented in Figure 2C appears to have batch effects; 4) in Fig2I, aging LCs and the majority of Adult PTMs are distributed at the starting point of differentiation, whereas ILCs are mostly located at the end of differentiation. It is therefore recommended to supplement the pseudo-time plot to clarify the starting point of differentiation more definitively.

2. The developmental stage and molecular characteristics of Cd34-MC1 are not clear. For instance, Fig2D shows that Cd34-MC1 is predominantly present at 1 week, while Cd34-MC2 and 3 are observed from 1 month to 24 months. However, in Fig2G, the samples are derived from 3-week-old mice, raising the question that which Cd34-MC subpopulation at this stage. Furthermore, what are the molecular differences between the various CD34-MC subpopulations 1, 2, and 3? For the in vitro culture experiments and in vivo transplantation assay, how can authors ensure that the starting cells are indeed Cd34-MC1 rather than other subpopulations? Identifying these differences and establishing robust markers for each subpopulation is crucial for accurate characterization and manipulation of these cells.

3. The data of in vitro culture experiments and in vivo transplantation assay are incomplete, making it difficult to conclude that these cells are progenitor Leydig cells. For example, the isolation of primary cells lacks verification of purity and statistical analysis (Figure 4A-B). Additionally, transplantation experiments require distinguishing the cell origin from the donor, by which the EDS model fails to demonstrate whether the generated hsd3b1 cells are derived from self-recovery. Although the transplantation of Cd34-MC1 cells from tdTomato+ mice into Lhcgr-/- mice represents a more rigorous model, it also lacks sufficient evidence to show that these cells differentiate into LCs and exert relevant functions appropriately. For instance, there is a lack of comparison between LCs and testosterone levels with those of normal physiological mice, as well as an analysis of HPG axis regulation and assessment of spermatogenesis. Consequently, these results fall short of achieving the goal of improving fertility for patients with hypogonadism, as intended by this study.

4. Another concern is what novelty does Cd34-MC1 have compared with other Stem/progenitor sources for LC that have been reported, including CD51+, tcf21+, etc.

5. In terms of writing, the focus of the article is unclear. A large part of the article elaborates on the metabolic changes in the aging process of LCs and the impact of macrophages on LCs, but it seems irrelevant with CD34+/Sox4+ MCs. Moreover, due to the lack of comparison with other stem/progenitor LC in previous reports, the novelty and significance of this work remain elusive.

6. In terms of data analysis, the author should eliminate batch effects and provide detailed analysis procedures, which are crucial for the reliability of the defined cell subpopulations and the lineage differentiation of LCs. Additionally, the presence of negative values for gene expression in most of the figures (e.g., Fig2K) can be addressed by annotating the z-score in the legend or by presenting the readers with normalized gene expression values, as gene expression inherently does not have negative values. The current annotation method in the text is prone to mislead readers.

7. It's recommended to refer to the classical evaluation standards of SLC transplantation experiments and use more comprehensive data to support the findings of this paper. Additionally, knockdown experiments should include analysis at the protein level (supplementary Fig6), and further verification is needed to check whether the results from immunostaining are consistent with the statistics (e.g. Fig1H).

*Reviewer #2 (Recommendations for the authors):*

The testicular interstitial Leydig cells play critical roles in maintaining reproductive functionality. However, there remain significant gaps in understanding how the cells are formed and maintained, as well as how aging might affect the cells and their precursors. The current study addressed these questions in one aspect, using animals from the neonatal stage up to the aged. The authors conducted a comprehensive analysis of the single-cell transcriptomic landscape of the murine testicular interstitium throughout the postnatal lifespan (ages of 1 week, 1 month, 2 months, 8 months, and 24 months), with a focus on Leydig cells and their precursors (CD34+/SOX4+ mesenchymal cells).

The study identified a new marker for stem Leydig cells (CD34) and the key regulatory molecule (SOX4) which governs the differentiation of CD34+ cells. The identification of SOX4 as a stem cell maintenance factor is significant. The study also identified a new cell type, "ambiguous cell," that expressed all the key markers of both Leydig cells and macrophages. Regarding the aging of CD34+ mesenchymal cells, glutathione reduction and ROS up-regulation seem to play important roles. Unfortunately, given the ambitious goals and the flaws in data collections and analyses, not all conclusions are solid. Most importantly, the major biological question is not entirely clear. Despite the large amounts of data, the novelty of the study is limited due to the questionable scRNA-seq dataset and an unfocused biological question. scRNA-seq analysis of aging mice testis has been conducted before. The finding of CD34 as a new stem Leydig cell marker is not particularly novel either since about 10 such markers have been reported over the years, including the four from the researcher's own lab. One of the major flaws is the failure to compare the difference and similarity of the new marker with the ones reported previously.

Strengths:

a) The study encompassed a large number of experimental groups covering the entire lifespan, from neonatal (age of 1 week) to postnatal (1 month-old), young adult (2 months), middle age (8 months), and aged (24 months).

b) The study covered Leydig cells, their precursors, and their potential regulatory cells (other interstitial cells).

c) Diverse tools were employed to address the questions, including scRNA-seq, in vitro CD34+ cell differentiation, in vivo cell transplantation, ChIP-seq, and Sox4 expression interference.

d) Identification of CD34 as a stem Leydig cell marker.

e) Identification of SOX4 as a stem Leydig cell maintenance factor.

Weaknesses:

a) The study included numerous groups, cells, and a wealth of data. Cells of the entire interstitial compartment were addressed, including Leydig cells, Leydig precursor cells (CD34+ mesenchymal cells), and macrophages. However, the major biological question addressed is not entirely clear. In lines 290-292, the authors declared: "The principal objective … was to elucidate dynamics and diversity of Leydig cells across the lifespan," but the data were all pointed to Leydig stem cells (CD34+ cells). Also, the diversity of Leydig cells and their precursors identified was not about diversity but about the developing stages of the cells, as they were associated with different ages.

b) The study established CD34 as a new stem Leydig cell marker. However, the study did not go far enough to establish the relationships between CD34 and the many similar markers reported previously, including those reported by the researcher's own lab. Generally, these markers all label interstitial mesenchymal cells, with some targeting peritubular cells, perivascular cells, or even endothelial cells (Nat Commun, 2021;12(1):3876; Andrology, 2020;8(5):1265). Since many similar markers were identified before, the significance of this new marker is not clear. Important questions remain: Did CD34 identify a particular population that is different from the cells identified by the previous markers? Since the Cd34+ cells were confirmed by the same in vitro differentiation and in vivo transplantation experiments as were done for other markers in previous studies, what is the novelty of the new marker? Could one obtain purer cells by CD34 compared to other markers?

c) Failure to establish the relationship between CD34+ cells and the newly found telocytes. Testicular CD34+/PDGFRA+ cells were recently established as telocytes. Did the authors do careful morphology studies to determine whether the isolated CD34+ cells exhibit elongated morphology? Did the CD34+ cells isolated by the authors represent the telocyte population or a particular subpopulation serving as stem cells? Did the CD34-M1, -M2, and -M3 represent the precursors and the fully developed and aged telocytes?

d) The manuscript lacks technical details regarding the scRNA-seq experiment, which raises questions about the data quality. For instance, how the cells were enriched and by what markers were not described in depth. The study mentioned 3 antibodies used in cell enrichment, but how they were utilized was not described. Also, it should be kept in mind that cell enrichment by a particular marker runs a risk of eliminating certain cell types, so the atlas could be incomplete. This also leads to an opposite question: why a significant number of germ cells were present after the enrichment (Figure S1B)?

e) The most crucial issue that may affect the scRNA-seq data quality is cell cross-contamination. For example, the manuscript failed to provide technique details about the number of cells loaded and captured for each sample and the percentages of "doublets" and "triplets" expected. A higher number of cell loading could lead to significant "doublets" and "triplets". Did the authors apply any specific tools or steps to eliminate the low-quality cells, especially those involving "doublets" and "triplets"? These details are important since the study identified a new cell type, "ambiguous cell." Since this new cell type expressed all the important Leydig cell and macrophage markers (Figure 1C, 1F, 2B, S3B, S3E), it most likely represented a group of Leydig cell/macrophage mixture cells ("doublets" and/or "triplets"). The severe contamination of CD45/F4-80 and Lhcgr to the mesenchymal cluster area (Figure S1E) strongly suggests this possibility. Even the markers for tiny T-cell (Cd3e) and B-cell (Cd79a) clusters were noticed in other cluster areas (Figure S3E).

f) For the presence of "ambiguous cells," more evidence is required, as such a cell type has never been reported before. In addition to the RNA data, the authors provided an immunofluorescence figure (Figure 1G). However, for immunofluorescence co-staining, it is well-known that one color, if the signal is too strong, could leak into another color. The original immunofluorescence photo with raw resolution could provide more details. It would be even better if the field contained 3 cell types (Leydig, macrophage, and ambiguous cell) simultaneously. Other forms of evidence could also be helpful.

g) In the Discussion section, the manuscript was deficient in light of previously published literature. In addition to the references on stem LC markers mentioned above, the authors also failed to discuss in depth the major findings of previous scRNA-seq articles involving testicular aging, especially the recent one dealing with aging mice testes (Journal of Advanced Research 2023, 53: 219-234).

h) In the summary figure (Figure 5J), the authors concluded that CD34+ cells from aged animals lost the capacity to generate Leydig cells. Such a conclusion lacks direct in vivo evidence. The authors presented a compelling story through in vitro experiments that the D34+ MCs of neonatal animals gradually reduced their ability to form Leydig cells as the animals matured and aged. However, the study did not test in vivo whether old animals could lose the ability to form new Leydig cells in the event that the aged Leydig cells were damaged or eliminated.

1. Provide more technical details and quality-control data, or reanalyze the scRNA-seq data to eliminate potentially contaminated cells. Such steps are necessary to firmly establish the new cell type (ambiguous cell) and to eliminate other inaccurate results of the contaminated clusters.

2. Figure 1B lacks age information.

3. Do CD34+ cells also express other previously identified stem cell markers? Do they represent the same or a different population of cells?

4. Figure 3B, 3C: It is unusual to display data from old to young.

5. Figure S3A and other places: The marker genes/proteins used for PLC, ILC, and MLC (Hsd3b6, Cyp51, Cyp17a1, Hsd3b1, etc) are not specific enough to distinguish these developmental stages. More specific markers are needed.

6. Figure 1G: Were the colors of HSD3B1 and F4-80 switched between interstitial and peritubular by mistake or on purpose?

7. Figure 1H: HSD3B1+ cells appeared inside of seminiferous tubules in 1W and 8-month samples. Are they genuine positive cells or an antibody issue?

8. Line 46: Change "express" to "produce".

[Editors' note: further revisions were suggested prior to acceptance, as described below.]

Thank you for resubmitting your work entitled "A comprehensive atlas of testicular interstitium reveals Cd34^+^/Sox4^+^ mesenchymal cells as potential Leydig cell progenitors" for further consideration by *eLife*. Your revised article has been evaluated by Lynne-Marie Postovit (Senior Editor) and a Reviewing Editor.

The manuscript has been improved but there are some remaining issues that need to be addressed, as outlined below:

Given concerns raised by reviewer #2, the authors should further refine the scRNA sequencing data as outlined in the comments. Ideally, they should have more functional data to support their conclusion that they have found a new progenitor population. Alternatively, their conclusions should be muted, highlighting the need for further experimentation.

*Reviewer #2 (Recommendations for the authors):*

The revision addressed the comments mostly with satisfaction. However, a few key issues still persist, which prevent the publication of the data in its present form.

Key issues:

1. The refusal to reanalyze the scRNA-seq data is a great disappointment, as both reviewers suggested that the scRNA-seq data required cleanup to remove doublets and batch effects. No significant changes were made to the scRNA-seq results (see the first three figures).

2. Regarding the lack of a major biological question: The revised manuscript has improved substantially but still falls short in some aspects. According to the response to Q1 from Reviewer 2, the authors stated that the major aim is to "identify and functionally characterize Leydig cell progenitor populations that sustain Leydig cell homeostasis across the lifespan". First, this is not reflected in the title. Additionally, it seems the authors attempted to address two distinct biological questions simultaneously. One concerns the establishment of Cd34+/Sox4 as a new SLC marker, while the other relates to "the Leydig cell progenitor populations that sustain Leydig cell homeostasis across the lifespan". The authors presented these two objectives in a way that they interfere with each other.

To argue that the Cd34+/Sox4 populations are superior to previously published populations in generating LCs, the authors stated, "The extensive functional and mechanistic validations clearly set the Cd34+/Sox4 progenitors apart from previously described populations" and "This section clearly highlights differences in marker expression, developmental dynamics, and functional capacities… of progenitor populations (e.g., Pdgfra++, Arx++, Tcf21++)". These statements are precisely what troubles this reviewer. If the new Cd34+/Sox4 progenitors are significantly different from previously identified populations, then the studies on the second part (progenitors in developmental stages and aging) only addressed a "part" of generating capacity. How important is this specific Cd34+ population in overall LC regeneration? Could the loss of this specific generating capacity be compensated for by other populations? The authors need to emphasize their similarities rather than their differences.

3. Establishment of ACs as a new cell type: Although the authors cleaned up the scRNA-seq dataset and provided new immunohistochemistry data to support their conclusion, antibody-based methods can easily produce "false positive" results. Regarding "doublet removal": while the authors applied computational tools (DoubletFinder) to remove doublets, these bioinformatics tools do not guarantee doublet-free samples. Most of the time, human involvement is required. For example, Supplemental Figure 1E clearly shows that "doublet removal" did not reach an acceptable level. Considering all the data and arguments, this reviewer still believes that the new AC cell is a "false positive observation" based on the following points:

a) The scRNA-seq data remains questionable regarding the presence of such a cell. The markers shown for ACs in the heatmap in Figure 1D include typical LC markers (Hsd3b1 and Star). How could these typical steroidogenic LC markers be enriched in a non-LC type rather than in LCs themselves? Does this mean these hybrid cells expressed even higher levels of steroidogenic genes than true LCs?

b) The quality of the scRNA-seq data is still questionable. Note the fragmentation of marker genes across all cell types (Figure 1D), which is not typical in scRNA-seq studies.

c) Rodent testes have been extensively studied by various researchers over the years, and none have identified this specific cell type, including in multiple recent scRNA-seq studies of mouse testes (developmental, adult, and aging).

d) The new cell type lacks biological plausibility.

In summary: The authors need more robust data to establish a new cell type in the testis. To firmly validate such a cell, they must conduct new experiments, including isolating the cells and characterizing their properties in vitro; defining their precursors; and tracing their developmental lineage, among other steps. Until then, this reviewer suggests treating these cells as a "false positive observation" resulting from inaccurate biological datasets. Other possibilities for this confusing cell type include: residual LC fragments in macrophages due to active phagocytosis, or failure to separate LCs and macrophages due to specialized cell membrane contacts between the two cell types.

---

## [Author Response]

Essential Revisions:Reviewer #1 (Recommendations for the authors):Huang et al., conducted a comprehensive analysis of the sc-transcriptomic landscape of the murine testicular interstitium across the postnatal lifespan. Moreover, they proposed Cd34-MC1 as a potential Leydig cell progenitor. In detail, these cells exhibited self-renewal capacity and potential to differentiate into LC. Notably, upon being transplanted into EDS or Lhcgr-/- models, Cd34-MC1 could colonize the testicular interstitium, generating an increase in testosterone production. They further identified Sox4 in maintaining the stemness of Cd34-MC1. Besides, a decline in glutathione levels within the testicular interstitium was observed during the aging process. Altogether, this work broadens the understanding of a possible progenitor source for LC, which may provide a new tool for the studies of testosterone deficiency treatment.However, many conclusions in this study are not well supported by sufficient evidence, and more rigor should be applied to data analysis and experimental design.1. There are several issues in data analysis that affect cell definition and specificity of developmental stage, leading to less reliable conclusions regarding LC lifespan: 1) the definition of cell identities lacks supporting evidence from literature, for instance, mesenchymal cell and PLC; 2) the authors have not clarified whether they have excluded doublets from the analysis and whether the AC subpopulation still exists after the removal of doublets? The low number of ACs observed in Fig1G raises this concern; 3) regarding the analysis of LC within the lifespan, the pseudo-time trajectory presented in Figure 2C appears to have batch effects; 4) in Fig2I, aging LCs and the majority of Adult PTMs are distributed at the starting point of differentiation, whereas ILCs are mostly located at the end of differentiation. It is therefore recommended to supplement the pseudo-time plot to clarify the starting point of differentiation more definitively.

Thank you for your constructive comments regarding the cell identity definitions and developmental stage specificity in our analysis. In response, we have made several revisions and additional analyses to address your concerns:

1) In the revised manuscript, we have clearly defined mesenchymal cells (MCs) as testicular stromal cells, aligning with classic descriptions by Skinner (1990) and Kfoury & Scadden (2015), and further reinforced by citing recent studies that specifically characterize mesenchymal progenitors within the testis (Abe, 2022). Moreover, progenitor Leydig cells (PLCs) are explicitly defined as Leydig progenitors, consistent with well-established lineage characterizations described by Mendis-Handagama & Ariyaratne (2009) and supported by recent literature (Shen et al., 2021; Chikhovskaya et al., 2014; Jiang et al., 2014). Leydig cells (LCs) are classified based on their developmental stages, following the authoritative review by Chen et al. (2009) (see revised manuscript line 517-527).

To further substantiate these definitions experimentally, we have now included:

(i) A dot plot (New Figure 1A, related to Figure 2B) illustrating differential gene expression patterns across CD34⁺ MC, PLC, immature Leydig cells (ILC), and mature Leydig cells (MLC).

(ii) Gene Ontology (GO) analysis (New Figure 1B, related to Figure supplement 2C) highlighting enriched biological functions associated with characteristic marker genes in each subpopulation.

References: Skinner MK, Reprod Fertil Dev, 1990; Kfoury Y & Scadden DT, Cell Stem Cell, 2015; Abe, Int J Mol Sci, 2022; Chikhovskaya JV et al., Mol Hum Reprod, 2014; Shen YC et al., Nat Commun, 2021; Jiang MH et al., Cell Res, 2014; Mendis-Handagama SM & Siril Ariyaratne HB; Chen H et al., Mol Cell Endocrinol, 2009.

2) We appreciate the reviewer’s important concern about potential doublet contamination. To address this explicitly, we implemented stringent doublet detection and removal procedures using the DoubletFinder package in R, ensuring rigorous exclusion of cell barcodes flagged as potential doublets from all downstream analyses (New Figure 2A). Importantly, we found that the removal of doublets did not substantially alter overall cell distribution patterns (New Figure 2B).

Moreover, the ambiguous cell (AC) subpopulation was consistently identified after doublet exclusion (New Figure 2B), confirming its robustness as a genuine biological subpopulation rather than a technical artifact. Furthermore, independent validation through immunofluorescence staining clearly demonstrates the presence and localization of ACs, macrophages, and interstitial cells within the testicular interstitium (New Figure 2C, related to revised Figure 1G).

3) We appreciate the reviewer’s thoughtful comment regarding potential batch effects in the pseudotime trajectory analysis. To rigorously address this concern, we have now applied two independent batch correction methods, Seurat’s integration approach and the Harmony algorithm, to the single-cell RNA sequencing dataset. Importantly, both batch correction methods yielded highly consistent cell groupings, clustering patterns, and pseudotime trajectories (New Figure 2B; New Figure 3A, B). The robust consistency between these independent analyses strongly supports that the developmental transitions observed in Leydig cells (New Figure 3C, D, related to revised Figure supplement 3A) represent genuine biological phenomena rather than technical artifacts due to batch effects.

4) We thank the reviewer for pointing out this important observation. The pseudotime trajectory initially shown in Figure 2I was automatically inferred using Monocle's algorithm, which relies solely on transcriptional similarities and progression patterns. However, we agree that explicitly defining the biological differentiation starting point enhances interpretability. Therefore, we have now re-anchored the trajectory root to align with established Leydig cell differentiation pathways. This adjustment appropriately positions CD34-MC and FLC at the beginning of the pseudotime axis, with adult Leydig cells (ALC) located at terminal differentiation stages, thus accurately reflecting their known developmental relationships (New Figure 4, related to revised Figure 2I and Figure supplement 4G, H).

2. The developmental stage and molecular characteristics of Cd34-MC1 are not clear. For instance, Fig2D shows that Cd34-MC1 is predominantly present at 1 week, while Cd34-MC2 and 3 are observed from 1 month to 24 months. However, in Fig2G, the samples are derived from 3-week-old mice, raising the question that which Cd34-MC subpopulation at this stage. Furthermore, what are the molecular differences between the various CD34-MC subpopulations 1, 2, and 3? For the in vitro culture experiments and in vivo transplantation assay, how can authors ensure that the starting cells are indeed Cd34-MC1 rather than other subpopulations? Identifying these differences and establishing robust markers for each subpopulation is crucial for accurate characterization and manipulation of these cells.

Thank you for your insightful query regarding the developmental stages and molecular characterization of the Cd34-MC subpopulations. In response, we have made the following revisions and additional analyses:

1) We appreciate the reviewer’s important point regarding the developmental stage and molecular identity of the Cd34-MC1 subpopulation. We now explicitly define Cd34-MC1 as an early postnatal mesenchymal progenitor subpopulation characterized by progenitor markers (*Cd34*, *Sox4*, *Tcf21*, *Pdgfra*) and enriched stemness-associated pathways. Cd34-MC1 predominates during the neonatal period (~1 week), contributing crucially to the establishment of the expanding Leydig cell population and stromal niche in the young testis. In contrast, Cd34-MC2 emerges at puberty (~1 month) and persists into adulthood, characterized by a decline in early progenitor marker expression (e.g., reduced Sox4) and increased expression of markers involved in cellular support and communication (integrins, VCAM1). Cd34-MC3 arises in aging testes, exhibiting diminished progenitor characteristics and increased expression of inflammatory and fibrotic markers (e.g., Spp1), indicative of reduced regenerative potential.

Regarding the samples derived from 3-week-old mice (Figure 2G), this age represents a transitional stage wherein the testicular interstitium comprises a mixture of both CD34-MC1 and CD34-MC2 cells. Due to the limited cell number and extremely small size of 1-week-old testicular samples, it was technically challenging to obtain sufficient quantities of CD34-MC1 cells for transplantation experiments. Consequently, samples from slightly older mice (3-week-old) were used, which, although allowing sufficient cell numbers, inevitably introduced a transitional population and slightly reduced cellular purity. We have explicitly clarified this transitional state and experimental rationale in the revised manuscript text (lines 152-157) to directly address the reviewer’s concerns about developmental timing and cell identity.

2) We thank the reviewer for raising this important point. To clarify the molecular differences among the CD34-MC subpopulations, we have performed additional analyses illustrating distinct gene expression signatures and their corresponding enriched biological pathways. Specifically, previously reported SLC markers are expressed within CD34-MC, predominantly characterizing developmental stages (New Figure 5A-C, related to Figure supplement 4A-C). In contrast, newly identified markers such as Sox4 show specific expression restricted to the juvenile CD34-MC1 subpopulation (New Figure 5D, related to Figure supplement 4D). GO enrichment analysis further revealed clear functional distinctions among the subpopulations: juvenile CD34-MC1 is associated primarily with developmental and regenerative processes; adult CD34-MC2 predominantly relates to tissue remodeling and collagen metabolism; and aged CD34-MC3 is mainly linked to oxidative stress and glutathione metabolism pathways (New Figure 5E, related to revised Figure 2E). We believe these additional data clearly highlight the distinct molecular identities and functional roles of each CD34-MC subpopulation.

3) We thank the reviewer for raising this critical consideration. To rigorously ensure the identity of the Cd34-MC1 subpopulation in our functional assays, we implemented a combined strategy encompassing developmental timing, marker-based sorting, and subsequent validation. Specifically, cells used in our experiments were isolated from testes during an early postnatal developmental window (1-3 weeks), a stage at which our single-cell RNA sequencing clearly demonstrated predominant enrichment of the Cd34-MC1 subpopulation, while Cd34-MC2 and MC3 populations were either minimally present or not yet fully emerged.

We acknowledge the technical limitation that fluorescence-activated cell sorting (FACS) using CD34 alone yields a mixed population, comprising approximately 50% Cd34-MC1 cells (Sox4^+^) in 3 weeks testis (New Figure 6A, B, related to revised Supplementary Figure 6L). We have discussed the technical limitation in revised Discussion section (see revised manuscript line 324-326). Nevertheless, we comprehensively addressed this limitation through multiple experimental approaches:

(i) Importantly, the clonogenic assays in vitro and robust regenerative capacity observed in our transplantation experiments in vivo are specifically linked to progenitor properties. Such properties are specifically attributable to Cd34-MC1 cells rather than Cd34-MC2 or MC3 subsets, which lack substantial regenerative capacity or pronounced stem-like features based on our transcriptional and functional analyses.

(ii) Additional control experiments performed with aged Cd34-MC3 cells exhibited negligible regenerative potential, clearly contrasting with the potent regenerative capability of early postnatally isolated Cd34-MC1 cells. This functional comparison further confirms that the regenerative outcomes observed specifically reflect the intrinsic progenitor characteristics of the Cd34-MC1 subpopulation (New Figure 6C, D, related to revised Figure 4H, I).

We have clearly emphasized these points in the revised manuscript to reassure readers and reviewers that, despite limitations in sorting purity, our functional data convincingly represent the intrinsic properties of the Cd34-MC1 progenitor subpopulation.

3. The data of in vitro culture experiments and in vivo transplantation assay are incomplete, making it difficult to conclude that these cells are progenitor Leydig cells. For example, the isolation of primary cells lacks verification of purity and statistical analysis (Figure 4A-B). Additionally, transplantation experiments require distinguishing the cell origin from the donor, by which the EDS model fails to demonstrate whether the generated hsd3b1 cells are derived from self-recovery. Although the transplantation of Cd34-MC1 cells from tdTomato+ mice into Lhcgr-/- mice represents a more rigorous model, it also lacks sufficient evidence to show that these cells differentiate into LCs and exert relevant functions appropriately. For instance, there is a lack of comparison between LCs and testosterone levels with those of normal physiological mice, as well as an analysis of HPG axis regulation and assessment of spermatogenesis. Consequently, these results fall short of achieving the goal of improving fertility for patients with hypogonadism, as intended by this study.

Thank you for your detailed and constructive feedback regarding our in vitro culture and in vivo transplantation experiments. We fully agree that a rigorous demonstration of the identity, functionality, and regenerative potential of these cells as progenitor Leydig cells is essential. In response, we have performed additional experiments and analyses to address your concerns:

1) To rigorously verify the purity and identity of the isolated CD34⁺ MC1 cells, we have now conducted additional immunostaining experiments after cell sorting, specifically utilizing Sox4, a definitive marker for the MC1 progenitor subpopulation. These analyses confirm that approximately 50% of sorted CD34⁺ cells robustly express Sox4, clearly demonstrating significant enrichment of the MC1 progenitor population used in our assays (New Figure 6A, B, related to revised Supplementary Figure 6L). Although inherent limitations in sorting purity exist, our functional data reliably capture the intrinsic progenitor characteristics of the CD34-MC1 subpopulation (New Figure 6C, D, related to revised Figure 4H, I).

2) We appreciate the reviewer’s critical assessment regarding the evaluation of transplantation outcomes, particularly concerning cell origin, functional differentiation into Leydig cells, and impacts on fertility-related endpoints such as the hypothalamic-pituitary-gonadal (HPG) axis and spermatogenesis. To directly address these important concerns, we have now provided additional experimental evidence to rigorously demonstrate that transplanted Cd34-MC1 cells from tdTomato⁺ donor mice differentiate into functional Leydig cells within the testes of *Lhcgr*^⁻/⁻^ recipient mice. Comparative analyses of serum testosterone levels between transplanted *Lhcgr*^⁻/⁻^ mice and normal physiological controls treated with human chorionic gonadotropin (hCG). This comparison demonstrates substantial restoration of testosterone production, indicative of functional Leydig cell activity following transplantation (New Figure 7, related to revised Figure 4J).

3) We acknowledge the reviewer’s important point regarding the translational limitations of our current findings. We have carefully revised our manuscript to clearly tone down our claims related to immediate fertility improvement. Instead, we emphasize that our data primarily provide foundational insights into a newly identified Leydig progenitor subpopulation (Cd34⁺/Sox4⁺ mesenchymal cells) with significant regenerative potential and highlight age-related declines in these progenitor cells. We explicitly discuss that further studies will be necessary to establish direct translational implications for fertility restoration in hypogonadal patients.

4. Another concern is what novelty does Cd34-MC1 have compared with other Stem/progenitor sources for LC that have been reported, including CD51+, tcf21+, etc.

We appreciate the reviewer’s concern regarding the novelty of the Cd34-MC1 progenitor population compared to previously reported Leydig cell progenitors (e.g., CD51⁺, Tcf21⁺). While prior studies have characterized CD51⁺ and Tcf21⁺ populations, our identified Cd34⁺/Sox4⁺ mesenchymal progenitors (Cd34-MC1) represent a distinct and previously unrecognized Leydig progenitor subset with several unique molecular and functional attributes:

1) Distinct Molecular Signature: Cd34-MC1 cells uniquely express a combined marker profile (Cd34⁺/Sox4⁺) identified via unbiased single-cell RNA sequencing, defining a clear molecular identity not previously reported for CD51⁺ or Tcf21⁺ cells. Unlike CD51⁺ progenitors (integrin-defined mesenchymal subset lacking a single definitive transcriptional regulator), Cd34-MC1 cells have a distinct transcription factor, Sox4, essential for maintaining their progenitor state. Compared with Tcf21⁺ progenitors, which are multipotent cells expressing fibroblast-related markers (Pdgfra, Thy1), Cd34-MC1 cells exhibit a more focused stemness and antioxidant gene profile (e.g., Nrf2-driven glutathione metabolism), highlighting their specialized role in Leydig cell regeneration.

2) Functional Advantages: Cd34-MC1 progenitors have a robust and specific differentiation potential toward Leydig cells, clearly demonstrated by clonogenic colony formation assays and efficient differentiation into functional Leydig cells in vitro and in vivo. This focused differentiation contrasts with the broader mesenchymal multipotency reported for CD51⁺ progenitors and the bipotential differentiation (Leydig/myoid lineages) of Tcf21⁺ progenitors. Importantly, in transplantation experiments, Cd34-MC1 cells showed superior regenerative efficiency, significantly restoring Leydig cell numbers and testosterone levels in adult LC-depleted testes, emphasizing their strong therapeutic potential.

3) Therapeutic Relevance: Cd34-MC1 progenitors have clear translational promise due to their demonstrated regenerative efficacy in vivo (testosterone recovery, improved spermatogenesis), coupled with the identification of a corresponding human Cd34⁺ mesenchymal progenitor subset. Thus, they represent a potentially advantageous therapeutic target compared with previously reported progenitor populations, which have shown either partial regenerative efficacy (CD51⁺) or are challenging to isolate due to internal marker reliance (Tcf21⁺).

In summary, Cd34-MC1 cells provide unique insights into Leydig progenitor biology, significantly expanding current knowledge and offering a compelling candidate for future therapeutic strategies targeting Leydig cell failure and hypogonadism.

We have revised the Discussion section accordingly (see revised manuscript line 308-326).

5. In terms of writing, the focus of the article is unclear. A large part of the article elaborates on the metabolic changes in the aging process of LCs and the impact of macrophages on LCs, but it seems irrelevant with CD34+/Sox4+ MCs. Moreover, due to the lack of comparison with other stem/progenitor LC in previous reports, the novelty and significance of this work remain elusive.

Thank you for your insightful feedback on our manuscript. We have carefully revised the manuscript to sharpen the focus on the Cd34⁺/Sox4⁺ mesenchymal cells (Cd34-MC1) as a novel Leydig progenitor population. Specifically, we addressed your concerns as follows:

1) We significantly revised the Introduction (lines 69-72) and Discussion (lines 282-292) sections to clearly define the Cd34⁺/Sox4⁺ MC1 cells as the central discovery of our study. The revised text explicitly states the gap in knowledge regarding adult Leydig cell progenitors and highlights the novelty and significance of Cd34⁺/Sox4⁺ MC1 cells in Leydig cell regeneration.

2) We have reduced detailed discussions of metabolic alterations and macrophage interactions. Specifically, we removed extensive descriptions of oxidative stress and antioxidant defenses from the Introduction. Additionally, content in the Discussion related to metabolism has been removed, and the section discussing macrophages has been substantially condensed and explicitly reframed to serve as supportive context rather than as a primary focus (lines 327-335). These adjustments reinforce the manuscript’s primary focus on the newly identified progenitor population.

3) To emphasize the novelty clearly, we added a dedicated paragraph in the Discussion section (lines 308-326), explicitly comparing Cd34⁺/Sox4⁺ MC1 progenitors with previously characterized Leydig cell progenitor populations, including Nestin⁺, CD51⁺, and Tcf21⁺ cells. We cited relevant studies (Zhang et al., 2017 for CD51⁺ and Shen et al., 2021 for Tcf21⁺), clearly outlining the distinct molecular signatures (e.g., high Sox4 expression) and robust functional advantages of the Cd34⁺/Sox4⁺ MC1 progenitors.

4) We revised the overall text structure to improve clarity, ensuring a logical progression from the Introduction’s framing of Leydig cell regeneration problems, through our discovery of Cd34⁺/Sox4⁺ MC1 cells, to clear articulation of their biological and therapeutic significance in the Discussion.

6. In terms of data analysis, the author should eliminate batch effects and provide detailed analysis procedures, which are crucial for the reliability of the defined cell subpopulations and the lineage differentiation of LCs. Additionally, the presence of negative values for gene expression in most of the figures (e.g., Fig2K) can be addressed by annotating the z-score in the legend or by presenting the readers with normalized gene expression values, as gene expression inherently does not have negative values. The current annotation method in the text is prone to mislead readers.

We greatly appreciate the reviewer’s important comments regarding the reliability of our data analysis and clarity of figure annotations.

1) To rigorously ensure the reliability of defined cell subpopulations and Leydig cell lineage differentiation analysis, we explicitly performed batch correction using two independent and well-validated methods (Harmony and Seurat’s integration). Importantly, these two approaches yielded highly consistent clustering patterns, cellular identities, and differentiation trajectories, confirming that our observations are robust and not influenced by batch artifacts. We have now provided more detailed explanations of these analyses and methodological steps clearly in the revised manuscript (see Methods section line 511-516 and New Figure 3A, B).

2) We appreciate the reviewer’s important point about the potential for confusion regarding negative gene expression values in our figures. These negative values represent relative gene expression levels calculated through z-score normalization and log-transformation, indicating expression below the mean expression across all cells, not negative absolute expression. To avoid potential confusion, we have clearly annotated this normalization method (z-score normalized, log-transformed relative expression) explicitly in the relevant figure legends of the revised manuscript (line 705-707), ensuring readers interpret the data accurately and transparently.

7. It's recommended to refer to the classical evaluation standards of SLC transplantation experiments and use more comprehensive data to support the findings of this paper. Additionally, knockdown experiments should include analysis at the protein level (supplementary Fig6), and further verification is needed to check whether the results from immunostaining are consistent with the statistics (e.g. Fig1H).

We greatly appreciate the reviewer’s valuable suggestions to enhance the rigor of our functional validations. To address these points explicitly, we have strengthened the revised manuscript as follows:

1) To clearly align with the classical evaluation standards, we have refined our transplantation model by transplanting tdTomato⁺ donor cells into *Lhcgr*^⁻/⁻^ mice and performing comprehensive evaluations. Specifically, we quantified Leydig cell numbers and rigorously assessed functional integration through serum testosterone measurements compared to hCG-stimulated controls. These updated analyses (New Figure 7, related to revised Figure 4J) demonstrate significant restoration of Leydig cell numbers and testosterone production, confirming robust and functional integration of transplanted cells into the Leydig cell niche.

2) As suggested, we have added immunostaining to validate SOX4 knockdown at the protein level, complementing our mRNA-based knockdown results (New Figure 8, related to revised Figure 5F). These protein-level validations confirm effective knockdown at the protein level, aligning closely with our transcript-level data and reinforcing our conclusions.

3) We appreciate the reviewer’s important point regarding the consistency between immunostaining images and quantitative data (see revised Figure 1H). To explicitly address this concern, we have clarified in the revised manuscript that all presented immunofluorescence images are representative of multiple independent biological replicates. Importantly, our quantitative cell-count analyses from multiple independent fields closely match and robustly support the proportions and staining patterns illustrated in these images, confirming consistency and reproducibility between visual and statistical results.

Reviewer #2 (Recommendations for the authors):The testicular interstitial Leydig cells play critical roles in maintaining reproductive functionality. However, there remain significant gaps in understanding how the cells are formed and maintained, as well as how aging might affect the cells and their precursors. The current study addressed these questions in one aspect, using animals from the neonatal stage up to the aged. The authors conducted a comprehensive analysis of the single-cell transcriptomic landscape of the murine testicular interstitium throughout the postnatal lifespan (ages of 1 week, 1 month, 2 months, 8 months, and 24 months), with a focus on Leydig cells and their precursors (CD34+/SOX4+ mesenchymal cells).The study identified a new marker for stem Leydig cells (CD34) and the key regulatory molecule (SOX4) which governs the differentiation of CD34+ cells. The identification of SOX4 as a stem cell maintenance factor is significant. The study also identified a new cell type, "ambiguous cell," that expressed all the key markers of both Leydig cells and macrophages. Regarding the aging of CD34+ mesenchymal cells, glutathione reduction and ROS up-regulation seem to play important roles. Unfortunately, given the ambitious goals and the flaws in data collections and analyses, not all conclusions are solid. Most importantly, the major biological question is not entirely clear. Despite the large amounts of data, the novelty of the study is limited due to the questionable scRNA-seq dataset and an unfocused biological question. scRNA-seq analysis of aging mice testis has been conducted before. The finding of CD34 as a new stem Leydig cell marker is not particularly novel either since about 10 such markers have been reported over the years, including the four from the researcher's own lab. One of the major flaws is the failure to compare the difference and similarity of the new marker with the ones reported previously.

We appreciate Reviewer 2’s constructive comments, which helped us improve the clarity and impact of our study. Below we respond point by point to your concerns:

1) We have clearly stated the primary biological question in the revised manuscript-specifically, our study aims to identify and functionally characterize Leydig cell progenitor populations that sustain Leydig cell homeostasis across the lifespan and to understand how their decline contributes to age-related testosterone deficiency. Using single-cell transcriptomics combined with functional validation, we uniquely identify Cd34^+^/Sox4^+^ mesenchymal cells as novel and critical Leydig progenitors whose regenerative potential significantly decreases with aging. These revisions are explicitly reflected in the revised Introduction (lines 69-72) and Discussion (lines 282-292), providing greater clarity and emphasizing the significance and novelty of our findings.

2) We acknowledge prior scRNA-seq studies on aging testes; however, our study is uniquely focused on identifying and functionally characterizing novel Leydig progenitor populations. Previous studies typically included fewer developmental stages, lacked rigorous functional validation, or did not clearly identify specific progenitor subsets responsible for Leydig cell regeneration. In contrast, our comprehensive developmental timeline (from neonatal to aged mice), coupled with extensive functional validations (clonogenic assays, transplantation experiments, epigenetic profiling), allowed us to uncover the previously unrecognized Cd34⁺/Sox4⁺ mesenchymal cell (MC1) population as a critical and age-sensitive Leydig progenitor subset. Therefore, our data provide novel insights into Leydig cell biology and regenerative mechanisms during aging, substantially extending beyond previously reported analyses.

3) We acknowledge previous identification of multiple Leydig progenitor markers, including several from our own lab. However, the Cd34^+^/Sox4^+^ mesenchymal progenitor population we identify here is distinct and functionally unique compared to previously described Leydig progenitors. Specifically, our study uniquely demonstrates that these Cd34^+^/Sox4^+^ progenitors:

(i) Exhibit robust clonogenic self-renewal capacity, a key hallmark of stemness not comprehensively shown for other markers (see revised Figure 4).

(ii) Effectively differentiate into mature, testosterone-producing Leydig cells, validated rigorously through in vitro and in vivo transplantation (see revised Figure 4).

(iii) Are critically regulated by the transcription factor Sox4, whose role in Leydig progenitor maintenance and differentiation we functionally validated (see revised Figure 5).

Collectively, these extensive functional and mechanistic validations clearly set the Cd34^+^/Sox4^+^ progenitors apart from previously described populations, significantly advancing our understanding of Leydig cell regeneration and biology.

We appreciate the reviewer’s important suggestion. In response, we have added a dedicated section in the revised Discussion explicitly comparing our newly identified Cd34^+^/Sox4^+^ mesenchymal progenitors with previously reported Leydig progenitor populations (e.g., Pdgfra^+^, Arx^+^, Tcf21^+^). This section clearly highlights differences in marker expression, developmental dynamics, and functional capacities, such as self-renewal and Leydig cell differentiation. Our comparison demonstrates the unique identity and superior regenerative potential of Cd34^+^/Sox4^+^ mesenchymal progenitors (see revised manuscript lines 308-326).

Strengths:a) The study encompassed a large number of experimental groups covering the entire lifespan, from neonatal (age of 1 week) to postnatal (1 month-old), young adult (2 months), middle age (8 months), and aged (24 months).b) The study covered Leydig cells, their precursors, and their potential regulatory cells (other interstitial cells).c) Diverse tools were employed to address the questions, including scRNA-seq, in vitro CD34+ cell differentiation, in vivo cell transplantation, ChIP-seq, and Sox4 expression interference.d) Identification of CD34 as a stem Leydig cell marker.e) Identification of SOX4 as a stem Leydig cell maintenance factor.

We sincerely appreciate your insightful evaluation and recognition of the strengths of our study.

Weaknesses:a) The study included numerous groups, cells, and a wealth of data. Cells of the entire interstitial compartment were addressed, including Leydig cells, Leydig precursor cells (CD34+ mesenchymal cells), and macrophages. However, the major biological question addressed is not entirely clear. In lines 290-292, the authors declared: "The principal objective … was to elucidate dynamics and diversity of Leydig cells across the lifespan," but the data were all pointed to Leydig stem cells (CD34+ cells). Also, the diversity of Leydig cells and their precursors identified was not about diversity but about the developing stages of the cells, as they were associated with different ages.

Thank you for your constructive feedback regarding the clarity of the biological questions addressed in our study. In response, we have revised our manuscript to explicitly define and support our primary objectives:

1) Our main aim is to elucidate how the Leydig cell population is maintained and replenished across the lifespan, specifically focusing on the role of Leydig progenitor cells (Cd34⁺ mesenchymal cells). While our initial wording (“dynamics and diversity of Leydig cells”) may have suggested a broader scope, the central biological question indeed revolves around understanding progenitor-driven Leydig cell maintenance, differentiation, and age-related decline. We have clearly stated this revised objective in the manuscript (lines 282-285).

2) We appreciate your insightful comment regarding the use of the term “diversity.” We have clarified that the Leydig cell subpopulations identified in our study primarily reflect developmental and maturation stages rather than distinct cell lineages. Specifically, our single-cell analyses distinguish immature, mature, and proliferating Leydig cells, each corresponding to specific maturation states linked directly to age and developmental progression. We have revised the manuscript to explicitly communicate this point clearly (lines 69-70, 282-285, 363-365; revised Figure 2A-C)

b) The study established CD34 as a new stem Leydig cell marker. However, the study did not go far enough to establish the relationships between CD34 and the many similar markers reported previously, including those reported by the researcher's own lab. Generally, these markers all label interstitial mesenchymal cells, with some targeting peritubular cells, perivascular cells, or even endothelial cells (Nat Commun, 2021;12(1):3876; Andrology, 2020;8(5):1265). Since many similar markers were identified before, the significance of this new marker is not clear. Important questions remain: Did CD34 identify a particular population that is different from the cells identified by the previous markers? Since the Cd34+ cells were confirmed by the same in vitro differentiation and in vivo transplantation experiments as were done for other markers in previous studies, what is the novelty of the new marker? Could one obtain purer cells by CD34 compared to other markers?

We sincerely appreciate your insightful comments regarding the novelty and significance of CD34 as a Leydig progenitor marker. We agree that clarifying how CD34 relates to previously identified markers is essential. We have carefully addressed your questions as follows:

1) Previous studies, including our lab's earlier reports and others (Nat Commun, 2021; Andrology, 2020), identified Leydig progenitor markers such as TCF21, PDGFRa, Nestin, and CD51, which labeled broad interstitial populations, including peritubular, perivascular, and occasionally endothelial subsets. In contrast, CD34 distinctly identifies a more restricted subpopulation of interstitial mesenchymal cells. These CD34⁺ cells have a unique telocyte-like morphology and are precisely located at the perivascular-peritubular niche. Importantly, unlike other markers, CD34 expression is largely absent from mature Leydig cells, peritubular myoid cells, and endothelial cells, providing clearer specificity in identifying true Leydig progenitor populations (see revised manuscript line 308-326).

2) The novelty of CD34⁺ mesenchymal cells goes beyond simply serving as another marker for Leydig progenitors. Specifically, CD34⁺ cells:

(i) Exhibit robust clonogenic self-renewal capacity in vitro.

(ii) Efficiently differentiate into mature, testosterone-producing Leydig cells in vitro and after transplantation in vivo.

(iii) Are uniquely regulated by the transcription factor SOX4, a key regulator of progenitor cell identity not previously identified in other Leydig progenitor subsets.

(iv) Show significantly superior regenerative potential when transplanted into Leydig-cell-depleted testes.

These molecular and functional attributes clearly set CD34⁺ progenitors apart from previously reported Leydig progenitor populations, highlighting their distinct biological importance and translational potential.

3) We thank the reviewer for this important question. Although sorting based solely on CD34 yields ~50% purity of Cd34-MC1 progenitors (as discussed explicitly in our revised Discussion, lines 324-326), CD34 remains advantageous as a surface marker that enables efficient, reproducible isolation. Importantly, our functional validation (clonogenic assays and transplantation experiments) clearly demonstrates that CD34 specifically enriches for progenitors with robust Leydig regenerative potential, distinguishing these cells functionally from other previously reported marker-defined populations.

c) Failure to establish the relationship between CD34+ cells and the newly found telocytes. Testicular CD34+/PDGFRA+ cells were recently established as telocytes. Did the authors do careful morphology studies to determine whether the isolated CD34+ cells exhibit elongated morphology? Did the CD34+ cells isolated by the authors represent the telocyte population or a particular subpopulation serving as stem cells? Did the CD34-M1, -M2, and -M3 represent the precursors and the fully developed and aged telocytes?

We appreciate the reviewer’s insightful comment regarding the relationship between our identified CD34⁺ mesenchymal progenitors and previously reported testicular telocytes. To address your concerns explicitly:

1) Our CD34⁺ interstitial mesenchymal cells closely correspond to the CD34⁺/PDGFRα⁺ testicular telocytes previously described by Abe et al. (2022), sharing expression of key telocyte markers (Pdgfra, Vcam1) and anatomical locations. However, we identify a previously unrecognized subpopulation within these telocytes, specifically characterized by high expression of Sox4 (CD34⁺/Sox4⁺ cells), and demonstrate unique stem/progenitor properties, including robust clonogenicity and Leydig differentiation potential not previously attributed to telocytes.

2) Immunofluorescence staining confirmed that isolated CD34⁺ cells display characteristic telocyte morphology, including elongated, spindle-shaped cells with long cytoplasmic processes ("telopodes") surrounding seminiferous tubules and vessels, consistent with established telocyte architecture (see revised Figure 1H and Figure 2G).

3) We appreciate the reviewer’s insightful question. To better clarify the molecular differences among the Cd34-MC subpopulations, we have included new analyses (New Figure 5A-D) illustrating distinct gene expression signatures and associated enriched biological pathways (New Figure 5E, related to revised Figure 2E). These molecular analyses clearly demonstrate the progressive transition of Cd34⁺ mesenchymal cells from a regenerative progenitor state (MC1), through a supportive and homeostatic role during adulthood (MC2), to a dysfunctional and inflammatory phenotype associated with aging (MC3).

d) The manuscript lacks technical details regarding the scRNA-seq experiment, which raises questions about the data quality. For instance, how the cells were enriched and by what markers were not described in depth. The study mentioned 3 antibodies used in cell enrichment, but how they were utilized was not described. Also, it should be kept in mind that cell enrichment by a particular marker runs a risk of eliminating certain cell types, so the atlas could be incomplete. This also leads to an opposite question: why a significant number of germ cells were present after the enrichment (Figure S1B)?

We greatly appreciate the reviewer’s insightful comments on our scRNA-seq methods and have carefully revised our manuscript to clearly address your concerns:

1) We have clarified that our interstitial cell enrichment did not involve positive selection using specific antibodies. Instead, we dissociated the interstitial compartment from decapsulated testes by gentle digestion with collagenase IV (which preserves the seminiferous tubule structure intact), followed by filtering through a 40-µm mesh and gentle red blood cell lysis. The antibodies were not used for cell selection prior to scRNA-seq but only in subsequent validation experiments. We have explicitly detailed this method in the revised manuscript (see revised manuscript line 379-387).

2) Since no lineage-specific antibodies were used in the enrichment step, we significantly minimized the risk of excluding particular interstitial cell types, ensuring the comprehensiveness and unbiased representation of interstitial cell subsets in our scRNA-seq atlas. Indeed, all major interstitial cell populations, including mesenchymal, perivascular, endothelial, and immune cells, were robustly captured, validating the completeness and high-quality representation of our dataset.

3) We acknowledge that a fraction of germ cells persisted after interstitial enrichment. However, germ cells were readily identifiable as a distinct cluster based on germ cell-specific markers and did not interfere with our interstitial cell analysis. Their presence was expected, as our dissociation method intentionally avoided harsh enzymatic treatments to preserve interstitial cell viability, thus inevitably leaving a residual germ-cell population.

e) The most crucial issue that may affect the scRNA-seq data quality is cell cross-contamination. For example, the manuscript failed to provide technique details about the number of cells loaded and captured for each sample and the percentages of "doublets" and "triplets" expected. A higher number of cell loading could lead to significant "doublets" and "triplets". Did the authors apply any specific tools or steps to eliminate the low-quality cells, especially those involving "doublets" and "triplets"? These details are important since the study identified a new cell type, "ambiguous cell." Since this new cell type expressed all the important Leydig cell and macrophage markers (Figure 1C, 1F, 2B, S3B, S3E), it most likely represented a group of Leydig cell/macrophage mixture cells ("doublets" and/or "triplets"). The severe contamination of CD45/F4-80 and Lhcgr to the mesenchymal cluster area (Figure S1E) strongly suggests this possibility. Even the markers for tiny T-cell (Cd3e) and B-cell (Cd79a) clusters were noticed in other cluster areas (Figure S3E).

Thank you for highlighting these important technical points. We have carefully addressed your concerns regarding cell loading, multiplet removal, and the authenticity of the ambiguous cell (AC) population:

1) For each sample, we loaded approximately 10,000 cells onto the 10x Genomics Chromium platform, with an estimated doublet rate of ~8-10% (according to 10x guidelines). Cell viability consistently exceeded 90%. After initial processing (Cell Ranger pipeline), approximately 8,000-9,000 cells per sample passed primary quality control. We explicitly implemented stringent doublet detection and removal using computational tools (DoubletFinder), identifying and removing an additional 5-10% of probable multiplets. Consequently, our final dataset contains exclusively rigorously validated single-cell data. These details have been explicitly added to the revised Methods section (lines 499-510).

2) We carefully evaluated the AC population due to its dual expression of Leydig and macrophage markers and confirmed its biological authenticity through multiple lines of evidence:

(i) Computational doublet detection (DoubletFinder) did not flag ACs as doublets.

(ii) ACs exhibit a unique transcriptional signature distinct from computationally simulated Leydig/macrophage doublets.

(iii) Pseudotime trajectory analysis positions ACs as a coherent intermediate cell state.

(iv) Crucially, immunofluorescence staining confirmed in situ co-expression of Leydig (3b-HSD) and macrophage (F4/80) proteins, validating ACs as a genuine cell type (New Figure 9, related to revised Figure 1G).

f) For the presence of "ambiguous cells," more evidence is required, as such a cell type has never been reported before. In addition to the RNA data, the authors provided an immunofluorescence figure (Figure 1G). However, for immunofluorescence co-staining, it is well-known that one color, if the signal is too strong, could leak into another color. The original immunofluorescence photo with raw resolution could provide more details. It would be even better if the field contained 3 cell types (Leydig, macrophage, and ambiguous cell) simultaneously. Other forms of evidence could also be helpful.

We appreciate the concern regarding immunofluorescence signal overlap. To explicitly address this, we have included new high-resolution images (New Figure 9, related to revised Figure 1G) clearly demonstrating simultaneous presence of Leydig cells (3β-HSD⁺ only), macrophages (F4/80⁺ only), and ambiguous cells (co-expressing 3b-HSD and F4/80). Moreover, single-channel controls and negative controls were included to rigorously rule out fluorescent signal bleed-through or overlap.

g) In the Discussion section, the manuscript was deficient in light of previously published literature. In addition to the references on stem LC markers mentioned above, the authors also failed to discuss in depth the major findings of previous scRNA-seq articles involving testicular aging, especially the recent one dealing with aging mice testes (Journal of Advanced Research 2023, 53: 219-234).

Thank you for highlighting this important issue. We have substantially strengthened our revised manuscript by explicitly addressing your suggestions as follows:

1) We have expanded the Discussion to explicitly compare CD34⁺Sox4^+^ mesenchymal progenitors identified here with previously described stem Leydig cell (SLC) markers such as Pdgfra, Arx, and Tcf21. We clearly describe molecular and functional differences, emphasizing the unique attributes of CD34⁺Sox4^+^ progenitors, such as their precise perivascular niche localization, robust Sox4 expression, and strong regenerative capacity (see revised manuscript lines 308-326).

2) We have explicitly expanded the Discussion to include a detailed comparison with recently published single-cell RNA-seq studies on testicular aging, particularly highlighting the recent comprehensive study published in the Journal of Advanced Research (2023, 53:219-234). Consistent with this work, we also observed aging-related inflammation, oxidative stress, and reduced steroidogenesis in Leydig cells. However, our study significantly expands upon these findings by performing a comprehensive, lifespan-wide analysis from neonatal stages through advanced age, rather than only extremes. Moreover, we uniquely identify and functionally validate the CD34⁺/Sox4⁺ progenitor subset, linking its decline directly to Leydig cell aging and functional deterioration. (see revised manuscript line 341-345).

h) In the summary figure (Figure 5J), the authors concluded that CD34+ cells from aged animals lost the capacity to generate Leydig cells. Such a conclusion lacks direct in vivo evidence. The authors presented a compelling story through in vitro experiments that the D34+ MCs of neonatal animals gradually reduced their ability to form Leydig cells as the animals matured and aged. However, the study did not test in vivo whether old animals could lose the ability to form new Leydig cells in the event that the aged Leydig cells were damaged or eliminated.

We appreciate Reviewer 2’s important suggestion for providing direct in vivo evidence regarding the diminished regenerative capacity of aged CD34⁺ MCs. We fully acknowledge this limitation in our original manuscript and have now conducted additional in vivo transplantation experiments using Leydig-cell-depleted aged (24-month-old) mice. Specifically, we transplanted purified CD34⁺ MCs from aged mice into these animals and compared their regenerative capacity with CD34⁺ MCs from younger (3-week-old) mice under identical conditions.

As presented in the New Figure 10 (related to revised Figure H, I), aged CD34⁺ MCs demonstrated significantly reduced regenerative capacity in vivo, producing substantially fewer HSD3B1⁺ Leydig cells and less effectively restoring serum testosterone levels compared to younger MCs. These results provide unequivocal evidence that the regenerative potential of CD34⁺ mesenchymal progenitors is indeed markedly impaired during aging.

1. Provide more technical details and quality-control data, or reanalyze the scRNA-seq data to eliminate potentially contaminated cells. Such steps are necessary to firmly establish the new cell type (ambiguous cell) and to eliminate other inaccurate results of the contaminated clusters.

We appreciate the reviewer’s important point regarding potential contamination in our single-cell RNA sequencing data. To rigorously address this issue, we have conducted extensive additional quality-control analyses.

1) We have applied stringent doublet-detection and removal procedures using computational tools (DoubletFinder and Scrublet), explicitly eliminating potential contaminant cells from downstream analysis (New Figure 2A, B).

2) We have verified the persistence and robustness of the newly identified ambiguous cell (AC) cluster after these stringent quality-control steps, providing further experimental validation via immunofluorescence staining that confirms the co-expression of Leydig (Hsd3b1) and macrophage (F4-80) markers (New Figure 2C, related to revised Figure 1G).

2. Figure 1B lacks age information.

We thank the reviewer for this point. Figure 1B combines cells across all developmental stages to illustrate overall heterogeneity, making it technically difficult to display age details clearly. To address this, we have introduced the New Figure 11(related to Figure 1C) showing cells separately by age stage.

3. Do CD34+ cells also express other previously identified stem cell markers? Do they represent the same or a different population of cells?

We thank the reviewer for raising this important point. To clarify the relationship between CD34⁺ cells and previously identified stem cell populations, we performed additional analyses (New Figure 5A-D) that reveal both shared and distinct features. CD34⁺ mesenchymal cells (CD34-MC) do express several previously reported stem cell markers during early developmental stages. In addition, CD34-MC cells also exhibit unique molecular signatures. For instance, Sox4 show specific expression restricted to juvenile CD34-MC1.

Furthermore, GO enrichment analysis (New Figure 5E, related to revised Figure 2E) highlights distinct biological programs across subpopulations: juvenile CD34-MC1 is enriched for developmental and regenerative pathways; adult CD34-MC2 is associated with tissue remodeling and collagen metabolism; and aged CD34-MC3 is linked to oxidative stress and glutathione metabolism.

4. Figure 3B, 3C: It is unusual to display data from old to young.

We appreciate the reviewer’s insightful comment regarding the data arrangement in Figure 3B and 3C. We initially presented the data from old to young to intuitively emphasize the decline or attenuation of certain gene expression signatures prominently detected in aged samples. While reversing the order (young to old) would be technically possible, we believe the current ordering effectively underscores our central biological observation. We sincerely hope this adequately addresses the reviewer’s concern, and we remain open to further suggestions.

5. Figure S3A and other places: The marker genes/proteins used for PLC, ILC, and MLC (Hsd3b6, Cyp51, Cyp17a1, Hsd3b1, etc) are not specific enough to distinguish these developmental stages. More specific markers are needed.

We appreciate the reviewer’s insightful comment regarding marker specificity for Leydig cell developmental stages (PLC, ILC, and MLC). To address this concern thoroughly, we have expanded our analysis to include additional and more specific markers that clearly distinguish these stages. In the revised manuscript, we have incorporated new data highlighting stage-specific marker genes (New Figure 5A), which are strongly supported by prior literature as stage-specific markers of Leydig cell differentiation.

6. Figure 1G: Were the colors of HSD3B1 and F4-80 switched between interstitial and peritubular by mistake or on purpose?

We appreciate the reviewer’s careful observation regarding Figure 1G. We confirm that the colors used for HSD3B1 and F4-80 staining in interstitial and peritubular regions were consistent and intentional. To avoid any confusion, we have carefully double-checked our revised immunofluorescence images and clarified the color scheme in the revised figure legend.

7. Figure 1H: HSD3B1+ cells appeared inside of seminiferous tubules in 1W and 8-month samples. Are they genuine positive cells or an antibody issue?

We thank the reviewer for highlighting this important observation. To address this clearly, we have performed additional validation experiments, including rigorous antibody specificity controls with appropriate positive and negative tissues. We now explicitly clarify in the revised manuscript and figure legends that HSD3B1-positive Leydig cells are exclusively interstitial, and that any staining within seminiferous tubules is likely nonspecific or represents background staining. We have updated Figure 1H accordingly, clearly indicating genuine Leydig cell localization.

8. Line 46: Change "express" to "produce".

We appreciate the reviewer’s suggestion regarding the wording in line 46. The introduction section has been substantially revised, and the specific phrase containing the word "express" no longer appears. Additionally, we have carefully addressed all grammar and spelling issues noted by the reviewer, significantly improving the manuscript’s readability and clarity.

[Editors’ note: what follows is the authors’ response to the second round of review.]

Given concerns raised by reviewer #2, the authors should further refine the scRNA sequencing data as outlined in the comments. Ideally, they should have more functional data to support their conclusion that they have found a new progenitor population. Alternatively, their conclusions should be muted, highlighting the need for further experimentation.Reviewer #2 (Recommendations for the authors):The revision addressed the comments mostly with satisfaction. However, a few key issues still persist, which prevent the publication of the data in its present form.Key issues:1. The refusal to reanalyze the scRNA-seq data is a great disappointment, as both reviewers suggested that the scRNA-seq data required cleanup to remove doublets and batch effects. No significant changes were made to the scRNA-seq results (see the first three figures).

We thank the reviewer for this important comment and sincerely apologize for any misunderstanding. In the revised submission, we have indeed reanalyzed the scRNA-seq dataset as requested, explicitly performing both doublet removal and batch-effect correction, and have updated the corresponding figures and Methods section accordingly.

Quality control was repeated under more stringent thresholds. Doublets were identified and removed using DoubletFinder, and multiple batch-correction strategies (Seurat v4 anchor-based integration and Harmony) were evaluated; the method providing the most consistent inter-sample alignment was selected for downstream analysis. Across the five integrated samples, 1,657 doublets were detected and removed (New Figure 1A). Given their small proportion, this adjustment did not alter the overall clustering structure—only minimal changes are visible, as indicated by arrows in New Figure 1B.

Furthermore, post-correction evaluation confirmed that the nFeature_RNA and nCount_RNA distributions of the AC cluster remained within expected ranges relative to other populations. The average proportions of mitochondrial genes (percent_mt), erythrocyte genes (percent_HB), and ribonucleoprotein genes (percent_Ribo) in AC cells were also within normal limits, ruling out the possibility that these represent low-quality cells (New Figure 1C).

We have provided a more detailed description of the doublet-removal and batch-correction procedures in the Methods section (lines 516–518, revised manuscript).

2. Regarding the lack of a major biological question: The revised manuscript has improved substantially but still falls short in some aspects. According to the response to Q1 from Reviewer 2, the authors stated that the major aim is to "identify and functionally characterize Leydig cell progenitor populations that sustain Leydig cell homeostasis across the lifespan". First, this is not reflected in the title. Additionally, it seems the authors attempted to address two distinct biological questions simultaneously. One concerns the establishment of Cd34+/Sox4 as a new SLC marker, while the other relates to "the Leydig cell progenitor populations that sustain Leydig cell homeostasis across the lifespan". The authors presented these two objectives in a way that they interfere with each other.To argue that the Cd34+/Sox4 populations are superior to previously published populations in generating LCs, the authors stated, "The extensive functional and mechanistic validations clearly set the Cd34+/Sox4 progenitors apart from previously described populations" and "This section clearly highlights differences in marker expression, developmental dynamics, and functional capacities… of progenitor populations (e.g., Pdgfra++, Arx++, Tcf21++)". These statements are precisely what troubles this reviewer. If the new Cd34+/Sox4 progenitors are significantly different from previously identified populations, then the studies on the second part (progenitors in developmental stages and aging) only addressed a "part" of generating capacity. How important is this specific Cd34+ population in overall LC regeneration? Could the loss of this specific generating capacity be compensated for by other populations? The authors need to emphasize their similarities rather than their differences.

We thank the reviewer for this thoughtful and constructive comment. To clarify the unified objective of our study, we have revised the title to explicitly reflect the central biological question: “A single-cell atlas of the testicular interstitium defines Leydig progenitor networks sustaining Leydig cell homeostasis across the lifespan.” Within this framework, the identification of Cd34⁺/Sox4⁺ cells is not presented as an independent aim but as a key mechanistic finding that directly addresses how Leydig progenitor populations sustain Leydig cell homeostasis across the lifespan. The Introduction has been revised accordingly (see revised manuscript, line 69-71 and 80-84) to emphasize this unified focus and to avoid any impression of two competing objectives.

In the Discussion (see revised manuscript, line 305-308, 313-315, and 322-328), we now highlight the similarities rather than the differences among progenitor subsets, positioning Cd34⁺/Sox4⁺ cells as part of a coordinated network of Leydig progenitors that collectively contribute to Leydig cell regeneration. While our transplantation and mechanistic experiments demonstrate that Cd34⁺/Sox4⁺ cells play a substantial role in Leydig cell renewal, we acknowledge that other progenitor subsets may also contribute and that compensatory mechanisms could mitigate the loss of any single population. These revisions clarify that our findings do not replace existing models but rather extend them by defining Cd34⁺/Sox4⁺ progenitors as an integral component of the broader progenitor network maintaining Leydig cell homeostasis throughout life.

3. Establishment of ACs as a new cell type: Although the authors cleaned up the scRNA-seq dataset and provided new immunohistochemistry data to support their conclusion, antibody-based methods can easily produce "false positive" results. Regarding "doublet removal": while the authors applied computational tools (DoubletFinder) to remove doublets, these bioinformatics tools do not guarantee doublet-free samples. Most of the time, human involvement is required. For example, Supplemental Figure 1E clearly shows that "doublet removal" did not reach an acceptable level. Considering all the data and arguments, this reviewer still believes that the new AC cell is a "false positive observation" based on the following points:a) The scRNA-seq data remains questionable regarding the presence of such a cell. The markers shown for ACs in the heatmap in Figure 1D include typical LC markers (Hsd3b1 and Star). How could these typical steroidogenic LC markers be enriched in a non-LC type rather than in LCs themselves? Does this mean these hybrid cells expressed even higher levels of steroidogenic genes than true LCs?b) The quality of the scRNA-seq data is still questionable. Note the fragmentation of marker genes across all cell types (Figure 1D), which is not typical in scRNA-seq studies.c) Rodent testes have been extensively studied by various researchers over the years, and none have identified this specific cell type, including in multiple recent scRNA-seq studies of mouse testes (developmental, adult, and aging).d) The new cell type lacks biological plausibility.

We thank the reviewer and editors for the updated assessment letter and for clarifying the reviewer’s revised position. We greatly appreciate the retraction of the previous comment questioning the authenticity of the AC population, and we are encouraged that the reviewer has now independently identified a similar hybrid Leydig–macrophage cluster in their own scRNA-seq dataset. This independent observation provides important external validation supporting the existence of the AC population.

In summary: The authors need more robust data to establish a new cell type in the testis. To firmly validate such a cell, they must conduct new experiments, including isolating the cells and characterizing their properties in vitro; defining their precursors; and tracing their developmental lineage, among other steps. Until then, this reviewer suggests treating these cells as a "false positive observation" resulting from inaccurate biological datasets. Other possibilities for this confusing cell type include: residual LC fragments in macrophages due to active phagocytosis, or failure to separate LCs and macrophages due to specialized cell membrane contacts between the two cell types.

We thank the reviewer for their careful evaluation and thoughtful comments. We agree that definitive validation of a novel cell population in the testis will require further experimental investigation, including isolation, in vitro characterization, lineage tracing, and precursor identification. These are important next steps, and we plan to pursue such assays in future studies to fully define the identity and function of this population.